

# Pressure Based Lift Estimation and its Application to Feedforward Load Control employing Trailing Edge Flaps

Sirko Bartholomay[1], Tom T.B. Wester[2], Sebastian Perez-Becker[1], Simon Konze[3], Christian Menzel[1], Michael Hölling[2], Axel Spickenheuer[3], Joachim Peinke[2], Christian N. Nayeri[1], Oliver P. Paschereit[1], and Kilian Oberleithner[1]

[1]TU Berlin, Institute for Experimental Fluid Dynamics and Technical Acoustics, Berlin Germany
[2]University of Oldenburg, ForWind - Institute of Physics, Oldenburg, Germany
[3]Leibniz Institute for Polymer Research Dresden, Germany

**Correspondence:** Sirko Bartholomay (sirko.bartholomay@tu-berlin.de, www.flow.tu-berlin.de)

**Abstract.** This experimental load control study presents results of an active trailing edge flap feedforward controller for wind turbine applications. The controller input is derived from pressure based lift estimation methods that rely either on a quasi-steady method, based on a three-hole probe, or on an unsteady method that is based on three selected surface pressure ports. Furthermore, a standard feedback controller, based on force balance measurements, is compared to the feedforward control. A Clark-Y airfoil is employed for the wing that is equipped with a trailing edge flap of $x/c = 30\%$ chordwise extension. Inflow disturbances are created by a two-dimensional active grid. The Reynolds number is $Re = 290,000$ and reduced frequencies of $k = 0.07$ up to $k = 0.32$ are analyzed. Within the first part of the paper, the lift estimation methods are compared. The surface pressure based method shows generally more accurate results whereas the three-hole probe estimate overpredicts the lift amplitudes with increasing frequencies. Nonetheless, employing the latter as input to the feedforward controller is more promising as a beneficial phase lead is introduced by this method. A successful load alleviation was achieved up to reduced frequencies of $k = 0.192$.

## 1 Introduction

Wind energy has become one of the most important sources in the energy mix, contributing $15\%$ of the consumed electricity in the European Union in 2019 (Komusanac et al. (2020)). One important challenge for this technology is to keep the cost of energy low. One of the key cost drivers is the material needed for the different turbine components. The amount of material used is largely determined by the loads that act on a turbine. Modern advanced controllers aim at reducing the load level a wind turbine is exposed to during operation so that less material can be used and the costs are lowered. Today, blade pitch is the most common actuator used in these advanced controller strategies, in particular cyclic and individual pitch control. However, the disadvantage of pitch actuators is their high rotational inertia, leading to a fairly high response time. Therefore, a large number of research projects have considered locally distributed active flow control devices within the last two decades. For example Pechlivanoglou et al. (2011) and Marrant and van Holten (2006) compared different technologies, such as trailing edge flaps,



micro-tabs, camber and twist control. Within the latter study, trailing edge flaps show the highest potential for the reduction of fluctuating loads.

Trailing edge flaps are not a new technology. They were already extensively analyzed in aircraft and helicopter industry. In the former, they are generally used for two goals: Firstly, reducing vibrations and thereby lowering fatigue (e.g. Arnold and

Dempster (1969)) and secondly, increasing the flutter speed (e.g. Sandford et al. (1975)). Besides fatigue load reduction (e.g. Leminos and Smith (1972)), research in the field of helicopters aimed at replacing swashplates by trailing edge flaps (e.g. Shen and Chopra (2004) and Thornburgh et al. (2014))

Extensive research into trailing edge flaps was also conducted in various wind energy research groups. Numerical two dimensional studies were for example carried out by Basualdo (2005) and by Buhl et al. (2005). One of the first 3D numerical

studies on trailing edge flaps based on BEM calculations, showed a reduction of $60\%$ in equivalent loads. Adding inertia to the flaps, noise to the measurements and delay to the processing reduced the effectiveness of the flaps, indicating the importance of structural lightness, measurement quality and fast processing (Andersen et al. (2006)).

A two dimensional experimental study was conducted by Bak et al. (2007), employing a piezo actuated trailing edge flap in prescribed motion to counteract airfoil-pitching-caused lift fluctuation. Velte et al. (2012) conducted an experiment on a two

dimensional wing that was exposed to sinusoidal disturbances. A feedback controller, based on lift estimation employing 64 pressure ports, was used. The lift reduction potential was shown for a reduced frequency of up to $k = 0.054$.

Castaignet et al. (2014) conducted a first full-scale test with active trailing edge flaps on a turbine with a rotor diameter of 27 m. Employing model predictive control, a load alleviation of $14\%$ was achieved. Within this experiment only 38 min could be measured, comparing 2 min of active control with the same time of unactuated measurements. Thereby no reproducible

inflow conditions were used.

These two facts, namely high costs of full-scale testing and limited inflow information, or reproducible inflow conditions, motivate model scale experiments. Heinze and Karpel (2006) and Bernhammer et al. (2013) present results of two-dimensional numerical and experimental studies analyzing piezo-actuated free floating flaps. This concept aims at high actuation bandwidth and large flap deflections. Consecutively, this concept was demonstrated on a research scale turbine by Navalkar et al. (2016).

Free floating flaps certainly have advantages as a high frequent actuation is possible. Yet, introducing a free-floating flap lowers the flutter speed and is therefore not fail safe. Bartholomay et al. (2018) recently performed experiments on the Berlin research scale turbine (BeRT) employing trailing edge flaps for different test cases. Herein, conventional closed loop control was used employing a PID controller.

The question arises on which sensor input should be used for active flow control devices (Cooperman and Martinez (2015)).

The most advanced technology might be hub mounted LIDAR (laser imaging, detection, and ranging) measurements that sense the incoming wind field. Yet, these systems are still comparably expensive and measurements of flow field asymmetries are inaccurate (Iribas et al. (2015)). Furthermore, Iribas et al. (2015) describe that the advantage of flow field forecasting of such a technique is less important for locally distributed flow control devices as their frequency bandwidth is comparably high.

Hence, other inputs are needed, preferably in-rotor-plane sensors which are likely to be installed anyway. For example, blade

root strain gage sensor are the obvious choice, when flapwise bending moments shall be minimized. However, these sensors





only measure a change in bending moment when a fluctuating load is already acting, limiting the load reduction potential of a strategy based on these sensors. Consequently, it is worth looking into distributed sensors, such as pressure based lift estimation methods. Herein, two methods are promising: blade mounted multi-hole probes or surface pressure methods. For example, Petersen et al. (2017) demonstrated within a full-scale test that on-blade-mounted five-hole probes are advantageous

devices to correlate inflow wind speed measurements to flapwise bending loads and to power production. They argue that this method is superior to met mast or lidar measurements as they are spatially closer to the turbine and less affected by the rotor than nacelle mounted anemometers. Yet, five-hole-probe measurements are corrupted due to local circulation. In order to correct for this, Petersen et al. (2015, 2017) propose a CFD based lookup table between the measured angle and the undisturbed angle of attack. Bartholomay et al. (2017) recently developed a comparable correction method based on XFOIL (Drela and

Youngren (2001)) calculations which requires substantially less computational ressources than CFD calculations. The method was validated against a URANS solver by Klein et al. (2017) for angle of attack and velocity measurements.

However, exposed blade mounted sensors such as five-hole probes are not practical on real world turbines, as they are likely to get clogged by insects, sand etc., might complicate maintenance procedures and their measurement might be corrupted by vibrations (Cooperman and Martinez (2015)). Therefore, a second approach for lift estimation, based on three surface pressure

ports, is developed within this study. The method is based on a study by Gaunaa and Andersen (2009) that was used in a two-dimensional experimental study by Velte et al. (2012).

The present study aims at load control based on pressure based lift estimation. An active trailing edge flap is employed, as they are one of the most promising active flow control devices (Marrant and van Holten (2006)). The two-dimensional experiments are conducted at the Reynolds number and reduced frequency corresponding to the 75% spanwise position of the

Berlin research turbine for $1p$ disturbances at rated conditions. Furthermore, higher reduced frequencies are analyzed to show the limits of the lift estimation methods and the controller settings. The feedforward control is grounded on either three-hole probe or on surface pressure lift estimates. The former is adopted from Bartholomay et al. (2017) and extended for additional trailing edge flap motion. The latter is based on a study by Gaunaa and Andersen (2009) and is extended in the current study in order to estimate the inflow velocity.

The remainder of the paper is structured in the following way: the description of the experimental setup is presented in section 2, followed by the lift estimation methods in section 3, the controller design in section 4 and the metrics for comparison of load control methods are presented in section 5. The results of the lift estimation methods are given in section 6 and the load control results are shown in section 7. A nomenclature is given in Appendix A.

## 2 Experimental setup

The present study is based on the recently designed and installed model-scale wind turbine BeRT. The rotor blades of this turbine employ Clark-Y airfoils and one blade is equipped with actuated trailing edge flaps that have a chordwise extension of 30%. The current two-dimensional experiments are conducted at a Reynolds number of $Re = 290,000$ which corresponds to the 75% spanwise position of the BeRT at rated conditions. Lately, Bartholomay et al. (2017) developed an experimental





method to generate a locally constant inflow disturbance that the turbine is exposed to. Furthermore, Bartholomay et al. (2018) employed yaw situations as test cases for potential load alleviation. These two inflow disturbances create $1p$ variations in flapwise bending moment at a reduced frequency of $k = \pi \cdot c/U_{\mathrm{rel}} = 0.09$, whereas $c$ is the chord und $U_{\mathrm{rel}}$ is the relative velocity in reference to the $75\%$ spanwise position. The former mentioned disturbance results in in-phase changes of angle of attack (AoA) and velocity, the latter results in variations which are opposite in phase.

In order to get further insights into lift estimation and load alleviation, a two dimensional model wing was built for the current study. It is exposed to inflow disturbances at the same reduced frequency and Reynolds number as for previously conducted experiments on the BeRT. The experiments were conducted in the recently developed two-dimensional active grid wind tunnel at the University of Oldenburg, which can mimic fairly independent changes in inflow angle of attack and velocity. In this section, the wind tunnel and the wing with trailing edge flap are presented, followed by the analysis of uncertainty aspects.

## 2.1 Wind tunnel

The test section of the closed loop wind tunnel of the University of Oldenburg is shown Fig. 1. The outlet of the wind tunnel has a cross section of $1.0\,\mathrm{m} \times 0.8\,\mathrm{m}$ $(w \times h)$. The tunnel has a closed test section of $2.6\,\mathrm{m}$ length, which consists of plexiglas walls to enable optical access for PIV measurements. The maximum velocity of the wind tunnel is $50\,\mathrm{m} \cdot \mathrm{s}^{-1}$ in open configuration and $45\,\mathrm{m} \cdot \mathrm{s}^{-1}$ with an attached grid. During the measurements the inflow velocity was set to $u_{\infty} = 15\,\mathrm{m} \cdot \mathrm{s}^{-1}$, resulting in a Reynolds number of $Re = 290,000$ based on the chord length of the airfoil which corresponds to the BeRT $75\%$ spanwise position.

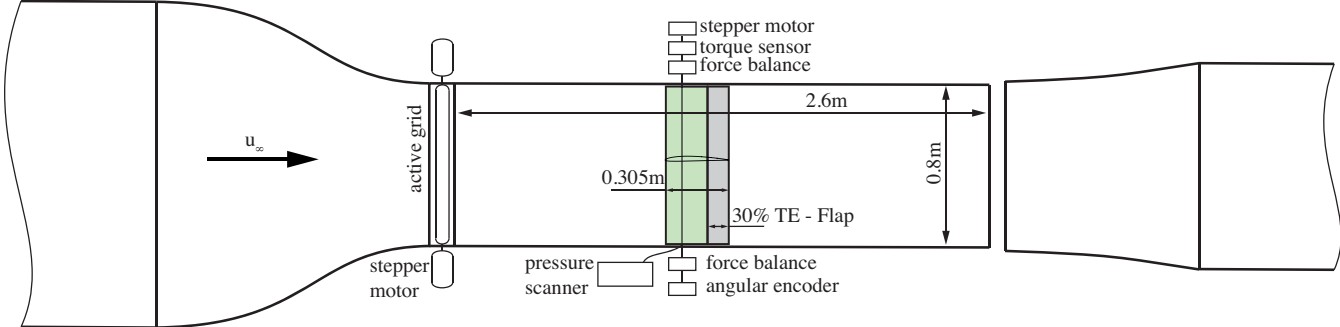

**Figure 1.** Wind tunnel setup with active grid, closed test section and installed 2D airfoil.

The inflow was modulated by an two-dimensional active grid (Wester et al. (2018)). The grid consists of nine individual movable vertical NACA 0016 airfoils with a chord length of $c_{\mathrm{grid}} = 71\mathrm{mm}$ and a span of $s_{grid} = 800\mathrm{mm}$. Every vertical axis was turned by a single stepper motor, which enabled individual movements of the axes and therefore the generation of very complex but two dimensional inflows. In the present study the seven inner axes performed a sinusoidal movement, whereas the two outer axes were used to increase or decrease the blockage. With this movement strategy a fairly sinusoidal angle of attack variation was generated by the inner axes and the outer axes additionally imposed a streamwise sinusoidal gust. By



changing the phase of the inner and outer axes the angle of attack and streamwise velocity fluctuations was either in-phase or out-of-phase.

The wing was mounted vertically on top of a turntable in the middle of the test section $1.1$m downstream of the active grid. The geometric angle of attack $\alpha_{\text{turntable}}$ of the airfoil was controlled by a stepper motor, which was located on top of the test section. $\alpha_{\text{turntable}}$ was kept constant during the measurements. Force measurements were realized by two force gauges of type $K3D120$ (*ME-Messsysteme*), one at the top and one on the bottom of the airfoil, and a torsion sensors type $TS110$ (*ME-Messsysteme*).

## 2.2 Wing with trailing edge flap

In reference to the BeRT experiment, a two-dimensional wing was designed (see Fig. 2). It is based on a Clark-Y airfoil that was slightly increased in thickness at the trailing edge for manufacturing purposes. The chord is $c = 0.305$m and the span is $s = 0.8$m. The shell of the main body was manufactured from glas-fibre-reinforced-plastic (GFRP) supported by an aluminium structure. A solid, 3d printed trailing edge flap with a chordwise extension of $30\%$ was hinged by a fiber reinforced polymer hinge.

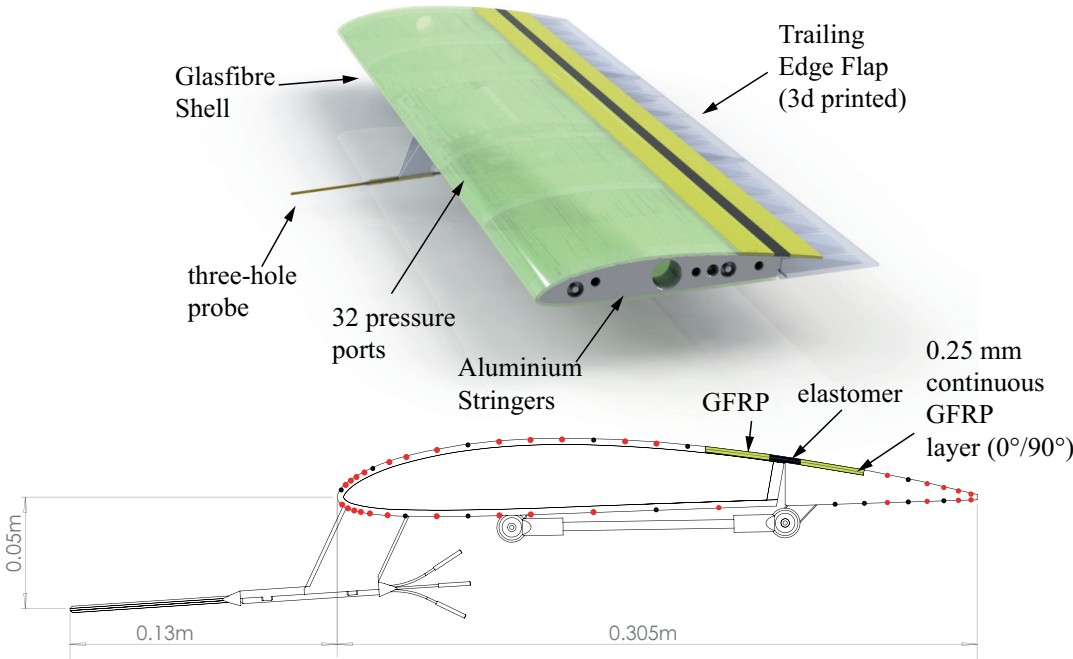

**Figure 2.** Sketch of the employed wing. Red dots on the surface indicate employed pressure ports, black dots show ports that were not used.

Concerning the flap material and its connection to the main body, different literature examples were analyzed. As found in a CFD study by Troldborg (2005), a rigid flap is aerodynamically slightly less efficient than a flexible curved flap. Hence, for





the same change in lift, a higher deflection angle is necessary. Furthermore, Buhl et al. (2005) indicated that an abrupt change of the surface geometry of an airfoil may lead to increased noise. However, a deflectable flap, made for example from rubber material, might show limits in fatigue life time on full-scale turbines, as their structural integrity might be compromised by environmental conditions, such as UV-light. Employing piezo actuators that bend the wing surface require a high voltage and

are therefore unlikely to be used on lightning exposed machines, such as wind turbines. Therefore, the current setup is based on a rigid flap, which could be build from GFRP in full-scale applications, and a continuous hinge. Thereby, a smooth surface on the suction side is achieved which ensures that the aerodynamic performance and the structural integrity of a potential full-scale application is not affected.

The hinge is made of woven glass fiber fabric, epoxy resin and elastomeric material and thus meets the lightning protection

requirements of potential full-scale applications, as no metallic parts are contained. The flexible zone in the mid section of the hinge contains a $0.25\mathrm{mm}$ thin layer of GFRP to improve the hinge structures in-plane stiffness and fatigue behavior, while still maintaining sufficient flexural freedom. Moreover, this thin GFRP layer provides a more homogeneous deflection curve of the joint area, compared to unreinforced elastomer. In order to prevent buckling or buckling-related damages of this thin laminate, both surface areas were covered by 0.5 mm thick elastomer. Within a bending fatigue experiment an equivalent hinge

was tested with over five million load cycles without showing any damage to the structure.

The flap was driven by a Faulhaber 32A servo motor located inside the wing. The connection to the trailing edge flap was realized by a lever arm located outside of the pressure side (Fig. 2).

The final setup is shown in Fig. 3. On the left side, the pressure side is seen with the servo motor, three-hole probe and surface pressure ports. Note, due to manufacturing limitations, not all pressure ports are in the same spanwise position. On the

right hand side in Fig. 3 the suction side is depicted. The view is from downstrem to upstream, thereby showing the inflow grid. Furthermore, the continuous hinge is shown as well as the trailing edge flap.

### 2.3   Measurement hardware

Measurements were conducted using a National Instruments cRIO 9068 data acquisition system employing three analog-in modules NI 9220. The acquisition frequency was set to $f_\mathrm{s} = 2$ kHz. The cRIO also sent the position commands to the servo

motor via Ethercat protocoll at an update frequency of $f_\mathrm{update} = 100$ Hz. The uncertainty of the sensors is summarized in Table 1.

The wing surface, including the trailing edge flap, was equipped with a multitude of pressure ports. During the measurement campaign 32 ports were measured, which are shown as red dots in Fig. 2. Furthermore, the wing was equipped with a custom-made three-hole probe (Fig. 3) that was previously used on the model wind turbine BeRT (Klein et al. (2017); Bartholomay

et al. (2018)). The two outer tubes have a $45°$ chamfer angle, whereas as the inner tube has a chamfer angle of $0°$. The probe was employed to obtain the angle of attack and the inflow velocity. The underlaying methodology is explained in section 3.1.

The three-hole probe as well as the surface pressure ports were connected by $0.5\mathrm{m}$ tubing to the sensors. The reference static pressure was used from two wall mounted pressure ports. These ports were located just upstream and in the center of the two



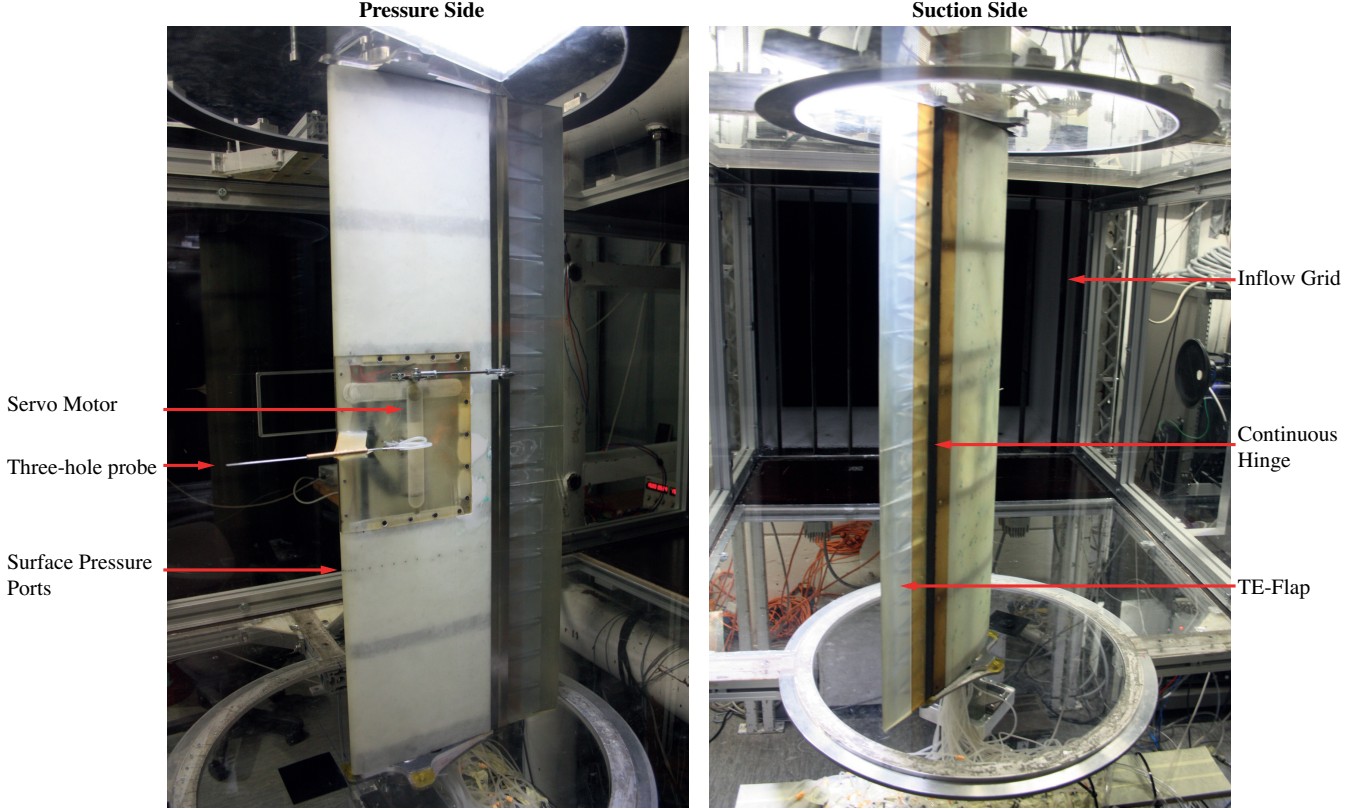

**Figure 3.** Mounted wing inside the wind tunnel. Left side: pressure side. Right Side: suction side, view from downstream to upstream.

**Table 1.** Summary of the measurement hardware uncertainty

| Device | Name | Uncertainty |
|---|---|---|
| Force Balance | $K3D120$ | 0.6 N |
| Torque Sensor | $TS110$ | 0.05 Nm |
| Angular Encoder | *Kübler Typ 5850* | $0.2°$ |
| Pressure Sensor | *HDOM010* | $+/-1$ Pa |
| Tubing Length | | 0.5m |
| Measurement Frequency | $f_s$ | 2 kHz |
| Controller Frequency | $f_{update}$ | 100 Hz |

most inner inflow axes of the active grid. This was necessary, as the original ring circuit for static pressure measurements was substantially affected by the changing adverse pressure created by the moving grid.





## 2.4 Frequency analysis of the test rig

The study presented in this paper is considered an aerodynamic and not an aeroelastic experiment. Therefore, the test-rig was analyzed for its structural eigenfrequencies. The excitation for this task was driven by the trailing edge flap. Multiple runs at various fixed frequencies were conducted and a time series of the force and torque balance were measured for each run.

Each time series was fourier transformed and the results were stacked, yielding a waterfall diagram. In Fig. 4 the result for the measurement of the moment is shown on the left, the normal force is depicted on the right side. The $y$-axis depicts the set actuation frequencies and in the $x$ axis corresponds to the fourier transform of the signal. The greyscale is representative for the amplitude. Diagonal lines represent the actuation frequency and its multiples, with the lowest line corresponding to the actuation frequency. As can be seen for the moment, there is a significant response at $16.6$Hz, which is expected to be the

torsional eigenfrequency of the test rig. This frequency appears also in the plot for the normal force. Additionally, a strong response can be seen at $24.8$Hz which is expected to correspond to the normal eigenfrequency.

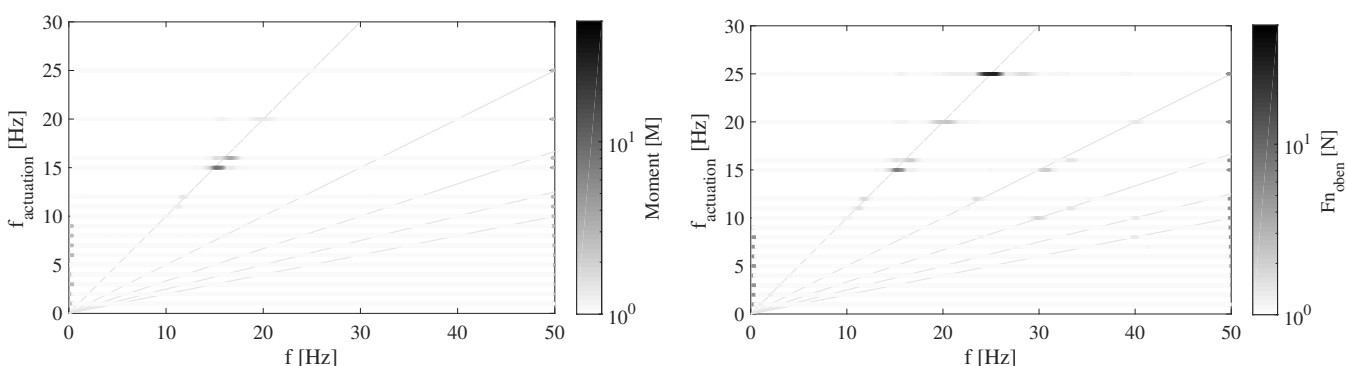

**Figure 4.** Waterfall diagram of the moment and normal force at different actuation frequencies for the trailing edge flap(y-axis). Experiment was conducted with the tunnel speed set to $u_\infty = 15$m/s

Furthermore, due to the acceleration of the flap inertia, a force is created which is opposite in sign to the aerodynamic lift created by the flap. Therefore, the force balance measures a zero lift amplitude where these two forces are equal. In the current setup this frequency is at 7 Hz.

## 2.5 Steady lift polars

Furthermore, the wing was tested for its baseline aerodynamic performance by measuring steady polar curves. For this purpose, the wing was turned by the turntable and the inflow grid was kept in its neutral position. Multiple repetitions with constant flap angles from $\beta = -10°$ to $\beta = +10°$ in steps of $\Delta\beta = 2.5°$ were conducted. The results of the measurements are depicted in Figure 5, where $C_L$ is shown in the most upper plot, $C_D$ in the middle and $C_M$ in the lower plot.

The slope of the lift polars in the linear range remains constant for all flap deflections (Fig. 5 upper plot). A positive flap deflection (flap turned towards pressure side) leads to a higher lift and vice versa. Furthermore, the angle of attack that corresponds to the lift maximum $\alpha(C_{Lmax})$ is reached, decreases when the flap angle is increased. Generally, the Clark-Y airfoil



shows a fairly constant level of lift beyond $\alpha(C_{\mathrm{Lmax}})$. At $\alpha = 22°$ all lift polars achieve almost the same lift, whereas beyond this point the differences increase again. However, the trend of higher lift due to higher positive flap deflection remains.

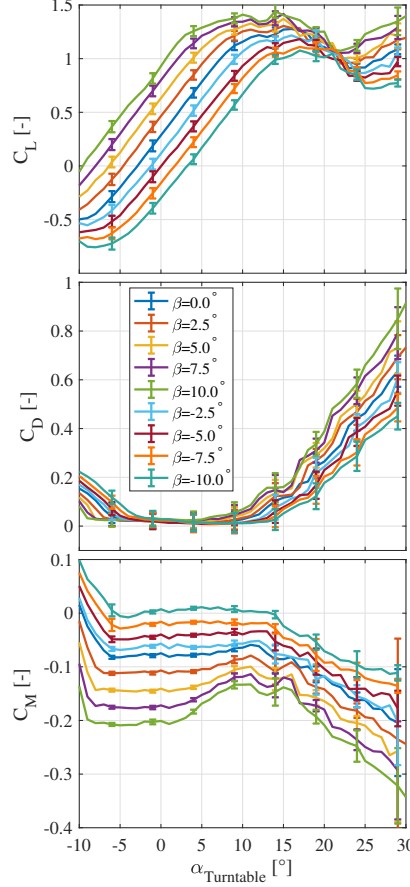

**Figure 5.** Steady polar curves. Measurements from force and torque balance measurements. Upper plot shows the lift, center plot the drag and lower plot the moment coefficient. Error bars correspond to the standard deviation.

The minimum drag created by the airfoil shows an almost constant level for all flap angle (see Fig. 5 middle plot). However, the range where this minimum angle is achieved is shifted to higher angles of attack with decreasing flap angles. Generally, larger flap deflections lead to higher drag with increasing AoAs. For lower AoAs the trend is inverse, with negative flap deflection leading to higher drag.

Furthermore, larger flap deflections lead to higher nose-up moments (smaller values of $C_{\mathrm{M}}$) as seen in the bottom plot of Fig. 5. Each of the curves shows a range where the moment coefficient is almost constant. However, this area is moved to higher angles of attack when the flap angle decreases. Moreover, the aforementioned range of constant moment covers larger angles of attack with decreasing flap angles.





As this study aims at load control to achieve constant lift, the lift slope in the linear region is most important ($dC_L/d\alpha = 0.092\ [1/°]$). Moreover, the flap effectiveness is calculated to $dC_L/d\beta = 0.067\ [1/°]$. These two values are used in pressure-based lift estimation methods to predict the current lift. Furthermore, the flap effectiveness is necessary to calculate the flap setpoint within the feedforward controllers.

## 3 Lift estimation methods

The current load control concept is based on pressure-based lift estimation. Therefore, the two employed methods are presented in this section. The first method relies on a three-hole probe and the second method on surface pressure ports.

### 3.1 Estimation procedure based on three-hole probe measurements

The first lift estimation method is based on a three-hole probe that is located $x/c = -0.43$ upstream of the leading edge. The measurement at this position is influenced by the induction of the blade. Therefore, a correction method based on quasi-steady assumptions to eliminate the induction effect was developed by Bartholomay et al. (2017) and compared to an URANS solution by Klein et al. (2017) for the rotating test-rig BeRT. This method is extended in the present study to include the change of induction for different trailing edge flap angles. A flowchart of the method is shown on the left side of Figure 6 and the corresponding sketch is given on the right side.

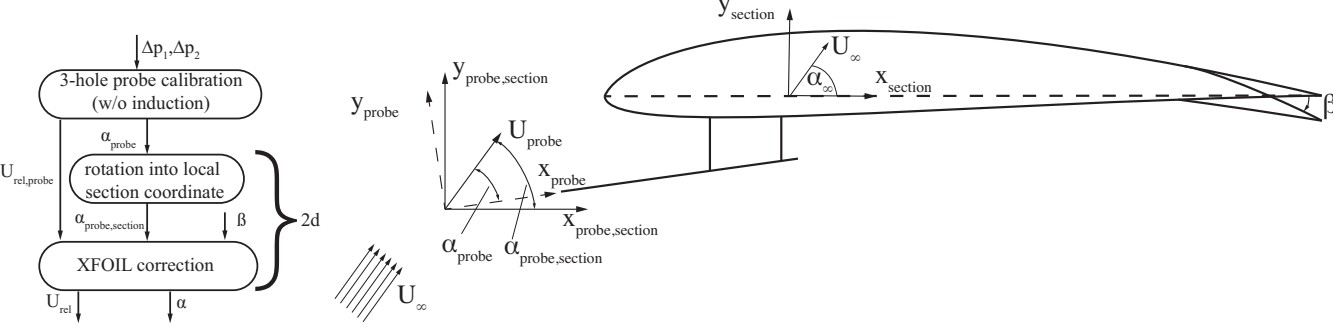

**Figure 6.** Flow chart and sketch for the estimation of angle of attack and velocity from three-hole probe measurements (adapted from Klein et al. (2017))

The three-hole probe lift estimate (3HP-estimate) relies on two pressure sensors that each measure the pressure difference ($\Delta p_1, \Delta p_2$) between one of the outer holes to the reference hole in the middle of the probe. The pressure differences were calibrated in a separate probe-only wind tunnel experiment, within an AoA range of $-30°$ to $30°$ for angular and velocity measurements. The calibration allows to derive the $\alpha_{\mathrm{probe}}$ and the velocity $U_{\mathrm{probe}}$ in the experiment, as seen in the flow chart on the left side in Fig. 6. In a second step, the angular difference between the probe and the chord of the wing is taken into account. Thereby, the flow angle $\alpha_{\mathrm{probe,section}}$ at the probe head is expressed in the section coordinate system. In the third step, XFOIL (Drela and Youngren (2001)) calculations are used in the current study to correlate the $\alpha_{\mathrm{probe,section}}$ and $U_{\mathrm{probe,section}}$ to





the uninfluenced $\alpha_\infty$ and $U_\infty$. These calculations are based on simulations from an AoA range of $\alpha = -30°$ to $30°$ in steps of $\Delta\alpha = 1°$ at flap angles of $-10°$ to $10°$ in steps of $\Delta\beta = 2.0°$. Values in between these predefined AoA and flap angles are linearly interpolated. This last step is comparable to the procedure presented by Petersen et al. (2015) that is based on 2d CFD simulations. However, it is expected to be considerably less computational expensive.

Generally, as this method is based on XFOIL, it is a quasi-steady approach for lift estimation. Nonetheless, for low frequent disturbances this method seems promising and will be evaluated for feed-forward control.

### 3.2    Estimation procedure based on three selected surface pressure ports

Due to their fragility, three-hole probes are of rather academic relevance and are unlikely to be used on industrial turbines. Therefore, a method is proposed that correlates three surface pressure measurements and the flap position to the angle of attack

and velocity. The method is based on unsteady thin airfoil theory (Gaunaa (2006)). Gaunaa and Andersen (2009) and Velte et al. (2012) apply the original work in order to estimate the unsteady lift based on the pressure difference of two pressure ports at $x/c = 12.5\%$ on the suction and pressure side. However, implicitly within this application of the theory, it is assumed that the local inflow velocity is known. On large turbines, the relative velocity can be estimated from the current rotational speed, hub height measurements of velocity, assumptions of the atmospheric boundary layer and the azimuthal position. However,

local flow differences can not be estimated from such a global approach.

Therefore, an approach is needed that estimates the local inflow velocity. This was done in other experimental studies before. For example Shipley et al. (1995) used a complete pressure port distribution around the airfoil to extract the velocity from the stagnation pressure. The stagnation point was found by searching only for the maximum pressure of the pressure distribution. The disadvantage of such a method is a high number of pressure ports that is necessary to achieve a good resolution at the

leading edge.

Thus, an approach is presented that employs one additional pressure port at the leading edge to estimate the inflow velocity. The remainder of this section presents the derivation of the inflow velocity first and secondly the derivation of the unsteady lift. A flow chart of the current method is given in Fig. 7.

In the current study, one single additional pressure port on the pressure side at $x/c = 0.6\%$ is used, in combination with the

pressure ports at $x/c = 12.5\%$ on the suction and pressure side, to estimate the inflow velocity. The approach is based on steady polar data, which are taken at the velocity of $U_\infty = 15 ms^{-1}$, and an AoA range of $\alpha = -8°...10°$ in steps of $\Delta\alpha = 1°$ and flap angles ranging from $\beta = -10°$ to $10°$ in steps of $\Delta\beta = 2.5°$. From the static polar data the normalized pressure difference is obtained:

$$\Delta c_{p,LE,SS12.5\%}(\alpha,\beta) = \frac{p_{LE}(\alpha,\beta,U_\infty) - p_{12.5\%c,SS}(\alpha,\beta,U_\infty)}{\frac{1}{2}\rho U_\infty^2} \tag{1}$$

and

$$\Delta c_{p,LE,PS12.5\%}(\alpha,\beta) = \frac{p_{LE}(\alpha,\beta,U_\infty) - p_{12.5\%c,PS}(\alpha,\beta,U_\infty)}{\frac{1}{2}\rho U_\infty^2} \tag{2}$$





Ignoring effects of changing Reynolds numbers, the dimensionless pressure differences are multiplied by the velocity range of $U_\infty = 10$ to $25 ms^{-1}$, yielding:

$$\Delta p_{p,LE,SS12.5\%}(\alpha,\beta,U_\infty) = \Delta c_{p,LE,SS12.5\%}(\alpha,\beta,15m/s)\frac{1}{2}\rho U_\infty^2 \tag{3}$$

and

5 $$\Delta p_{p,LE,PS12.5\%}(\alpha,\beta,U_\infty) = \Delta c_{p,LE,PS12.5\%}(\alpha,\beta,15m/s)\frac{1}{2}\rho U_\infty^2 \tag{4}$$

Thereby, a lookup table, given for a flap angle $\beta = 0°$ in Fig. B1, is created which relates the velocity to a function of the two pressure differences and the flap angle:

$$U_\infty = f(\Delta p_{p,LE,SS12.5\%}, \Delta p_{p,LE,PS12.5\%}, \beta) \tag{5}$$

During the measurements, the velocity estimation feeds directly into the procedure of the lift estimation shown in Fig. 7.

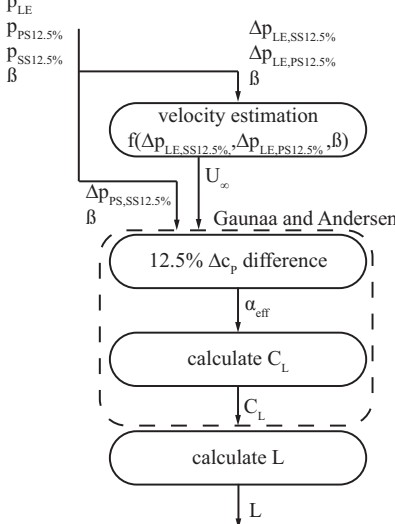

**Figure 7.** Flow chart for estimation of the unsteady lift based on selected surface pressure measurements.

In a second step, the unsteady lift is estimated based on the derivation of Gaunaa and Andersen (2009) and Velte et al. (2012). This method relies on the calculation of the effective AoA, which corresponds to the effective three quarter AoA containing the bound and shed circulation. The effective AoA is derived from the pressure difference between suction and pressure side, according to Gaunaa and Andersen (2009)

$$\frac{\Delta p(x)}{\frac{1}{2}\rho U_\infty^2} = g_c(x)\alpha_{\text{eff}} + g_{\text{camber}}(x) + g_\beta(x)\beta + g_{\dot{\alpha}}(x)\frac{\dot{\alpha}c}{U} + g_L(\ddot{\alpha},\dot{\beta},\ddot{\beta}), \tag{6}$$

where $g_c(x)$ corresponds to the circulatory pressure difference distribution, $g_{\text{camber}}(x)$ corresponds to the pressure difference coefficient due the camber of the airfoil and $g_\beta(x)$ represents the pressure difference coefficient due to the deflection of the





trailing edge flap. The last two terms in Eq. (6) are denoted by Gaunaa and Andersen (2009) as added mass effects. Note, these are not related to added mass effects due to airfoil, flap or flow motion. Yet, they signify the change on the pressure distribution that is introduced by the fact that the camber line is not straight but cambered and eventually further deflected through the current flap position. $g_{\dot{\alpha}}(x)$ is the pressure difference function due to pitching motion, which is shown from thin airfoil theory

to be zero at $x/c = 12.5\%$. The last term in Eq. (6) corresponds to higher order terms. An approximate pressure difference is found by neglecting the latter, using the pressure difference at $x/c = 12.5\%$ and neglecting the pitch rate term as done by Gaunaa and Andersen (2009), which yields:

$$\frac{\Delta p(x = 0.125c)}{\frac{1}{2}\rho U_\infty^2} = K_1 \alpha_{\text{eff}} + K_2 \beta + K_3. \tag{7}$$

The constants $K_1$, $K_2$ and $K_3$ are derived from the static polars and $\alpha_{\text{eff}}$ yields:

$$\alpha_{\text{eff}} = \frac{1}{K_1}\left(\frac{\Delta p(x = 0.125c)}{\frac{1}{2}\rho U^2} - K_2 \beta - K_3\right). \tag{8}$$

The effective angle of attack is then inserted into:

$$C_L = \underbrace{k_{\text{c}} \alpha_{\text{eff}}}_{\substack{circulatory \\ forces}} + \underbrace{k_{\dot{\alpha}} \frac{\dot{\alpha} c}{U_\infty}}_{added\ mass} + \underbrace{k_L(\ddot{\alpha}, \dot{\beta}, \ddot{\beta})}_{high\ order\ terms} \approx \frac{dC_L}{d\alpha}(\alpha_{\text{eff}} - \alpha_0), \tag{9}$$

where $k_c$ corresponds to the steady lift slope and $k_{\dot{\alpha}}$ to the added mass term due to airfoil motion. The latter is given as $k_{\dot{\alpha}} = \pi/2$ in Gaunaa and Andersen (2009). Following Velte et al. (2012) and Gaunaa and Andersen (2009), the lift coefficient

can be approximated by neglecting the added mass term. Thereby, an estimation of the unsteady lift is given, which does not include added mass effects due to changes of AoA or flap motion. Experimentally this method was employed by Velte et al. (2012) for a NACA 64418 airfoil with an active trailing edge flap. Finally, the lift is calculated from equation

$$L = \frac{1}{2}\rho U_\infty^2 cs(C_{\text{L}} + \frac{dC_{\text{L}}}{d\beta}\beta), \tag{10}$$

where the inflow velocity was taken from the estimate conducted in the first step.

**4  Controller design**

The presented pressure-based lift estimation methods are used in a static feedforward control approach to alleviate fluctuating loads. Feedforward control has the advantage, that controller action can be taken before the error is experienced by the system (Johan and Murray (2008)). In this particular case, the trailing edge flap can be actuated before the balance measures a change in lift. Therefore, the disturbance has to be measured earlier by an alternative sensor, pressure-based AoA and velocity mea-

surement in this case, and good process models -lift estimation- have to be present. The drawback of feedforward control is that model and measuring uncertainty cannot be controlled. In the present study, two feedforward controllers are compared to a PID based feedback controller.





For each controller setting the same low-pass filter setting is used. The cut-off frequency was set to $f_{\text{cut-off}} = 7$ Hz as the mechanical setup of the flap leads to negative lift responses beyond the cut-off frequency due to the acceleration of the flap inertia, as explained in section 2.4. The low-pass filter introduces a loss of gain and a phase shift that is expected to lower the capability of the setup. In order to keep the same filter setting for all controller types, a low-pass filter and not a band-pass filter was chosen. Furthermore, the update frequency of the controller loop is $f_{\text{update}} = 100$ Hz.

## 4.1 Feedback controller

In Figure 8, the feedback-controller is shown. It is important to note that the lift was taken from the force balance measurement. The reference lift is found by taking a $10$ s-mean from the current lift, without the disturbance being active. The PID constants were chosen as $K_p = 0.54$, $Ti = 0.009722$ min, $Td = 0$ using the standard Ziegler-Nichols method.

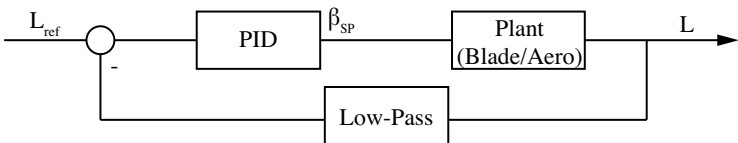

**Figure 8.** Feedback PID controller based on lift measurements of the force balance. Reference lift is based on a $10$ s mean at constant inflow conditions. $\beta_{\text{SP}}$ denotes the setpoint of the flap. The system plant comprises the flap mechanics and the aerodynamics.

## 4.2 Feedforward control based on the three-hole probe

The controller based on the three-hole probe, denoted 3HP-controller, is shown in Fig. 9. The first block from left indicates the calculation of the angle of attack and velocity as given in Fig. 6. Additionally, taking the current flap position into account, the lift is calculated. After low-pass filtering, the lift is subtracted from the reference, which is also taken from the 3HP-estimate of the lift. Note, for a fair comparison, the low-pass filter was chosen with the same settings as for the feedback controller. Finally, the flap setpoint $\beta_{\text{SP}}$ is determined by the use of the slope given by the static lift polars (see Fig. 5).

**Figure 9.** Feedforward controller based on three-hole probe lift estimate. The first two blocks corresponds to the lift estimation presented in section 3.1. The flap angle setpoint is calculated based on the error in reference to a $10$ s mean. The latter is also based on three-hole probe measurements.





### 4.3 Feedforward control based on three surface pressure ports

Figure 10 shows the second feed-forward controller (PP-controller), which is based on the surface pressure port lift estimate The only difference in comparison to the 3HP-controller consists of the lift estimate, which is based on the surface pressure ports described in section 3.2. Also for this controller, the same low-pass filter setting was chosen.

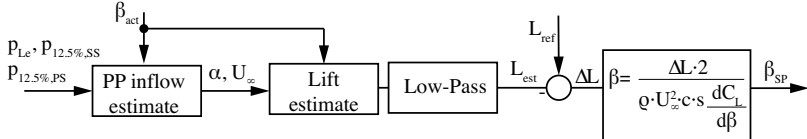

**Figure 10.** Feed-Forward controller based on surface pressure port measurements. The first two blocks corresponds to the lift estimation presented in section 3.2. The flap angle setpoint is calculated based on the error in reference to a 10 s mean. The latter is also based on surface pressure port measurements.

### 5 Metrics for comparison of load control devices

To allow for a consistent evaluation of each control method different metrics are employed. Plumley (2015) gives a detailed overview over different fatigue load metrics, starting with the standard deviation of time series, cumulative power spectral densities (CPSD) and (short term) damage equivalent loads (DELs). Whereas the standard deviation is a quick way to compare fluctuating loads, it does not give any information on important frequencies in the time series. Therefore, CPSD are useful, as they shed light on the frequencies that contribute most to the signals energy. Additionally, it is useful to calculate DELs as they provide a concise and representative number of the loading scenario. The latter two metrics are employed in the current paper and are therefore explained in the following two sections.

### 5.1 Cumulative power spectral density

The explanations in the current section follow Plumley (2015). Power spectral density plots are used to find frequencies which contain the most energy and thereby, where the most damage is created. The calculation in this paper is based on Welchs power spectral density which is then cumulated to highlight dominant components. These components can be seen as steps (see for example Fig. 17) which facilitates the comparison between the baseline and the controlled signals.

### 5.2 Short term damage equivalent loads

Damage equivalent loads are a standard metric in the wind energy industry. They are calculated in the time domain and take the material fatigue into account. In the current study only short term DELs are calculated in order to compare different time series to one another. The time series are low-pass filtered at a cut-off frequency of $f = 15$ Hz, which corresponds to the first eigenfrequency of the test rig, before DELs are calculated.





The calculation of DELs was performed using the NREL (2015) MLife code, which is publicly available. Hayman (2012) describes in the accompanying report the procedure to process the DELs, which is explained here briefly. The calculation is based on rainflow counting of load cycles in the time domain, resulting in cycle counts $n_i$ for different loading amplitudes $L_i$. Based on Miner's rule, which assumes linear accumulation of damage $d_i$ at different loading, the total damage is given as

$$D = \sum d_i = \sum \frac{n_i}{N_i(L_i)} = \frac{n_{eq}}{N_{eq}(\text{DEL})}, \tag{11}$$

where $N_i(L_i)$ describes the number of sinusoidal cycles to failure for each specific loading amplitude, which is found by employing Wöhler-S-N curves. Furthermore, the total damage calculated in Eq. (11) is expressed by one equivalent load (DEL) at a chosen number of cycles $n_{eq}$. In order to find the DEL, the Whöler exponent $m$ is needed, which is chosen $m = 10$ for composite material in the present study. The ultimate load $L_{ult}$ technically needed for Wöhler-S-N curves cancels out in the derivation. Thereby, equation 11 yields

$$\sum n_i \cdot L_i^m = n_{eq} \cdot \text{DEL}^m \tag{12}$$

In order to calculate the DEL, the equivalent frequency is chosen as $f_{eq} = 1 Hz$. Thereby, the equivalent number of cycles is found by $n_{eq} = f_{eq} \cdot T$, where $T$ is the elapsed time of the time series. By rearranging Eq. 12, the damage equivalent load is found:

$$\text{DEL} = \left( \frac{\sum n_i \cdot L_i^m}{n_{eq}} \right)^{(1/m)} \tag{13}$$

Summarizing, comparing signals for their success of load alleviation, it is important to look at different methods. Employing CPSDs are important to reveal what frequencies are affected, but they might falsely suggest a controller superiority, where there isn't. Therefore, DELs are necessary to quantify damage over all frequencies and to assess the effect on the component.

## 6   Results of the lift estimation methods

In order to assess the lift estimation methods, three scenarios with increasing complexity are analyzed: active flap motion with constant inflow, fixed flap with inflow disturbance and active flap with inflow disturbances. The aim is to assess the limits of the presented methods. Therefore, the reduced frequencies range up to fairly high values even though they might have limited interest for the wind turbine community.

### 6.1   Constant inflow conditions - active flap

The quality of lift estimation is analyzed first in test cases where the AoA is set to $\alpha_{turntable} = 5°$ and the inflow velocity is kept constant at $U_{\infty} = 15 m \cdot s^{-1}$. The flap executes a sinusoidal motion with an amplitude of $\beta = 5°$ and at a fixed frequency. The results of the analysis are shown in Fig. 11. It shows the phase and the gain of the complex transfer function between the flap setpoint signal and the lift measurements (balance and full surface pressure integration) or the presented lift estimation





methods. For the calculation of the gain a steady lift equivalent served as a reference. This is considered the lift that would be achieved, if the flap was to settle at the setpoint amplitude and the lift was to reach its corresponding steady value:

$$L_{\text{ref}}(f) = 1/2 \cdot \rho \cdot U_\infty^2 \cdot c \cdot s \cdot \left( C_L(\alpha = 5°) + \frac{dC_L}{d\beta} \cdot \beta_{SP}(f) \right) \tag{14}$$

The flap actual position (Fig. 11, black line), corresponds to the ratio between the amplitude of the actual flap amplitude to

the setpoint amplitude. Furthermore, $L_{\text{Balance}}$ corresponds to the lift force deduced from the balance measurements, $L_{\text{FULLPP}}$ to the calculated lift based on the integration of the lift over the complete set of pressure ports. $L_{3\text{HP}}$ and $L_{\text{PP}}$ show the result of the three-hole probe and three surface pressure port method, respectively. Additionally, the unsteady aerodynamic model ATEFlap (Bergami and Gaunaa (2012)) was used for comparison. In order to account for the non-zero thickness of the airfoil, the indicial constants for the model are calculated in reference to Bergami et al. (2013) and slightly adjusted: $A = [0.389; 0.264]$

and $b = [0.380; 0.0564]$. The deflection shape integrals are calculated for the flap with a chordwise extension of $30\%$ according to (Gaunaa, 2006, EQ. 39 / 40) and normalized by the half chord to $H_{y,i}^* = -0.0958$, $H_{dyd\epsilon,i} = -0.0358$ and $F_{dydxLE}^* = 0.142$.

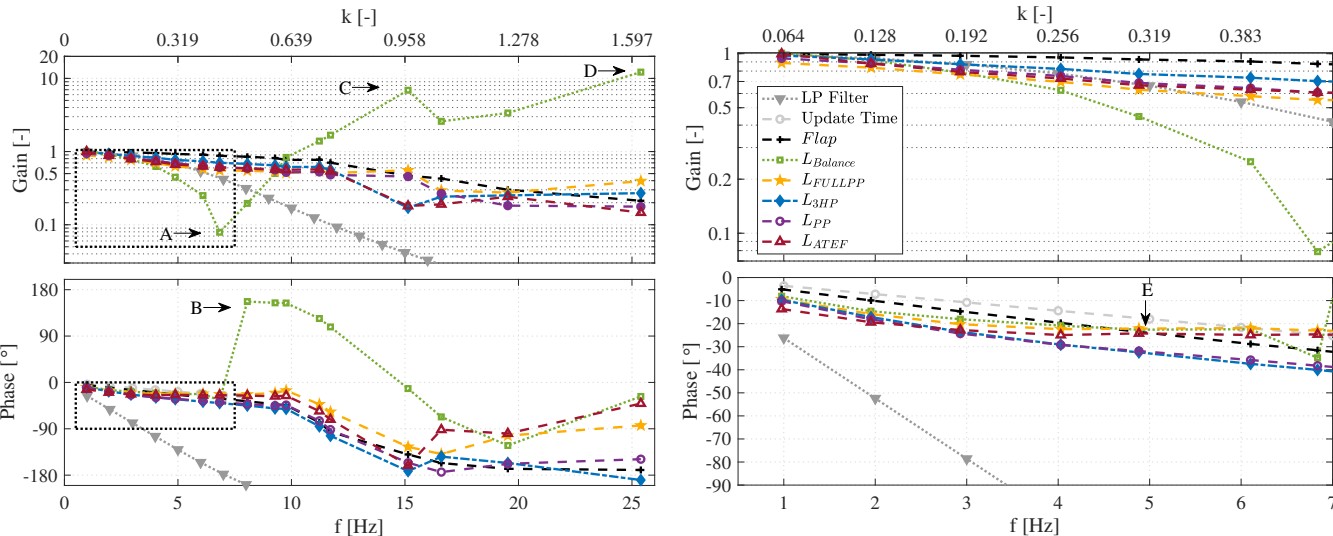

**Figure 11.** Bode plot of the transfer functions. Grey Lines correspond to the filter setting and update time of the controller loop. Black line: transfer function from flap setpoint $\beta_{\text{SP}}$ to flap actual position $\beta_{\text{actual}}$. All other lines correspond to $\beta_{\text{SP}}$ to the lift measurements or lift estimates. Left side: frequency range up to 26 Hz. Right side: Zoom in the frequency range up to 7 Hz.

Concerning the flap motion, it can be seen in the upper plots in Figure 11 (black line) that the flap follows its setpoint signal up to 10 Hz with little lost on the gain. Beyond this point, the amplitude decreases to $21\%$ at the frequency of $f = 26$ Hz and the phase drops to $\varphi = -180°$. This indicates that the present mechanical setup shows a decent functionality up to a frequency

of $f = 10$ Hz. At higher frequencies the introduced phase lag and the small gain poses a high challenge for load alleviation applications.

Considering the lift measurement of the balance $L_{\text{Balance}}$ (Figure 11 green line), it is seen at the marker *A* that the gain decreases to $8\%$ at $f = 7$ Hz. This is caused due to the acceleration of the flap inertia that creates a force amplitude which is





$180°$ phase shifted to the flap position. After the minimum is reached, the gain raises again, as the flap is accelerated faster with increasing frequency and thereby the force created by accelerated flap inertia dominates the lift response. At $f = 9Hz$ the gain of the other measurements is reached but with leading phase. The gain keeps further increasing with increasing frequency and in point $C$ and $D$ eigenfrequencies of the test-rig are hit, as shown in section 2.4. This leads to enormous gains, which are
not related to the flap motion only, nor are they representative for the acting aerodynamic forces.

Additionally, the lift is measured based on the integration of the complete pressure port set distributed on the surface of the wing (Fig. 11 yellow line - $L_{\text{FullPP}}$). It is seen that the lift does not drop as indicated by the balance measurements, which is plausible, as the flap inertia is not measured by the pressure ports. However, added mass effects are measured by the integrated lift and by the balance measurements, which is seen in point $E$ where the phase of the lift measurement becomes greater than
the phase of the flap actual position. This point is denoted phase turning point and will be discussed later in this section.

Concerning the phase of the three-hole probe (Fig. 11 blue line) and the surface pressure port estimate (purple line), it can be seen that up to a reduced frequency of $k = 0.19$ the phase difference to the integrated lift is small. However, beyond this point the phase error increases, as added mass effects are not taken into account in the lift estimations. Note, that in the case where only the flap is moving and no inflow disturbance is present, the lift estimates lag the integrated lift in phase. This is
expected due to the location of the probes ($x/c = -43\%$ for the three-hole probe and $x/c = 12.5\%$ for the pressure ports) and the fact that the change of circulation is created due to the trailing edge flap motion. This change in circulation alters the flow field around the airfoil, however, the change is introduced at the trailing edge. Thus, the three-hole-probe is affected lastly in reference to the other methods.

In order to understand the phase turning point seen at marker $E$ in Fig. 11, the lift hysteresis loops, $C_{\text{L}}$ over $\beta$ in this case,
are investigated in Fig. 12. The phase turning point becomes apparent by the change in direction of the loops from counter-clockwise to clockwise. Starting from the reduced frequency of $k = 0.065$ to $k = 0.130$ the counter-clockwise loop gets wider first, indicating the increasing effect of unsteady aerodynamics and hence, the wake memory effect that leads to a delayed lift response to the change of flap motion. However, beyond $k = 0.130$ the loop closes again and is virtually a straight line at $k = 0.324$. Increasing the frequencies further leads to a reopening of the loop, however, the direction of the loop changes
due to added mass effects. The effect of phase turning happens at the reduced frequency of $k = 0.318$ (Fig. 11 marker E), which is higher than the phase turning point of $k = 0.144$ as reported by Motta et al. (2015). However, the latter concerns an airfoil pitching case, where the accelerated mass is substantially larger in comparison to the current case where only the flap is moving. The balance measurements and integrated lift were compared to the ATEF model and the phase turning point agrees well between the two measurements and the model (Figure 11 marker E).

## 6.2  Disturbed inflow conditions with fixed and active flap

Within the second and third test case the wing was exposed to inflow disturbances at a mean AoA of $\alpha_{\text{turntable}} = 5°$ with fixed and active trailing edge flap motion.

Figure 13 shows time series of the lift when the wing is exposed to disturbed inflow conditions. On the left hand side, test cases are depicted, when the trailing edge flap is fixed in its neutral position. On the right hand side, the flap is active. The



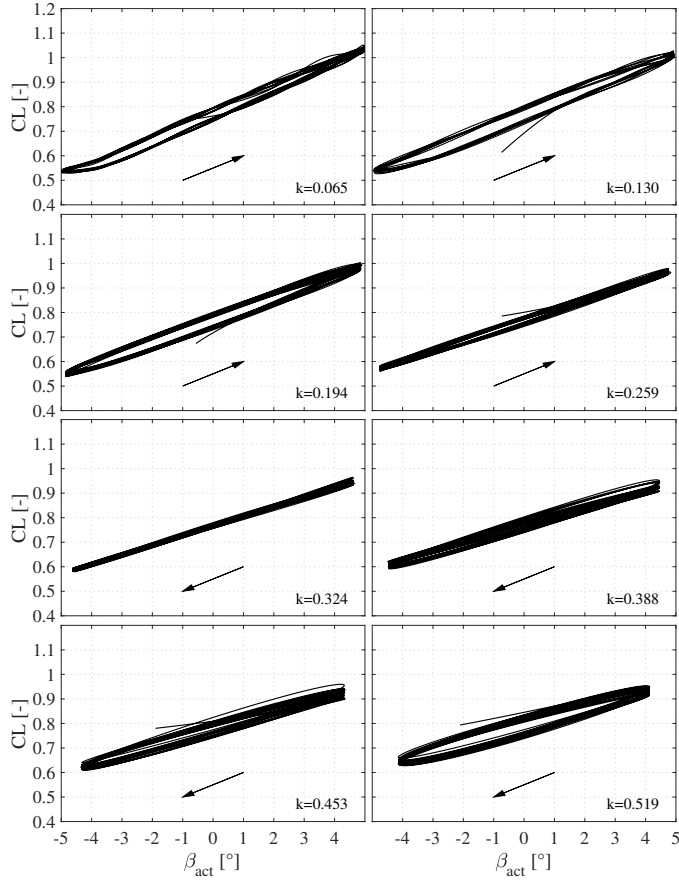

**Figure 12.** $C_L$ over actual flap angle for increasing frequencies. The arrow indicates the sense of rotation of the loop. Up to the frequency of $k = 0.259$ the sense is counter-clockwise, clockwise for higher frequencies.

different rows correspond to the different disturbance frequencies in ascending order. In each plot the comparison between the previously presented estimation methods to the lift measurement based on the integration of the complete set of pressure ports $L_{\text{FULLPP}}$ is given. As can be seen for the fixed flap cases (Fig. 13 left side), the comparison between the $L_{\text{FULLPP}}$ and the lift estimation based on three surface pressure ports $L_{\text{PP}}$ is fairly good up to a reduced frequency of $k = 0.256$. At the

5 highest frequency, $k = 0.319$, the lift amplitude is slightly overestimated. Considering the same estimation method, $L_{\text{PP}}$ for cases with actively moving flap (Figure 13 right side), the trend is generally kept, however, the differences are slightly larger in comparison to the $L_{\text{FULLPP}}$. Regarding the lift estimate based on the three-hole probe $L_{\text{3HP}}$, it can be seen that the lift amplitude is overestimated in all cases. Due to the lack of incorporation of unsteady effects into this method, this trend worsens with increasing frequencies. Furthermore, this method shows a significant phase lead in comparison to the full pressure result

10 $L_{\text{FULLPP}}$ due to the position of the probe. This phase lead is beneficial for load control as will be seen in section 7.



**Figure 13.** Comparison of the lift estimate to full pressure distribution at $\bar{\alpha} = 5°$. Left side: Inflow disturbances, fixed flap. Right side: Inflow disturbances and active flap.

In order to quantify the results of the lift estimation, the phase difference and the normalized root mean square deviation (NRMS) between the presented methods and fully integrated lift was calculated. The NRMS is given in the following equation:

$$\text{NRMS} = \frac{\sqrt{\overline{\left(L_{\text{FULLPP}} - L_{\text{est}}\right)^2}}}{\max(L_{\text{FULLPP}}) - \min(L_{\text{FULLPP}})}, \tag{15}$$

where the time average is used for the root mean square. Figure 14 shows the results of the comparison, where the upper graph
5   depicts the NRMS and the lower plot shows the phase difference. The latter is calculated by means of cross-correlation between the signals. The NRMS is calculated after alignment of the signals by the calculated phase difference.

Concentrating on the NRMS of the three-hole probe measurements for the fixed flap case first (see Fig. 14 upper plot). The deviation of this quantity stays fairly constant at about $10\%$ until a frequency of $f = 3Hz$ and starts increasing from that point towards $40\%$ at $f = 5$Hz. This high error is to be expected, as wake memory effects and lift changes due to non-circulatory





terms are neglected within this method. Considering an actively moving flap for the same case leads to an additional error of about 5%, as additional unsteady effects are present, which are not captured. Regarding the phase difference of the three-hole probe, a linear increase of the phase for the case of a fixed flap from $\varphi = 10°$ to $\varphi = 33°$ at a reduced frequency of $k = 0.319$ can be seen (see Fig. 14 lower plot). This increase corresponds to a fairly constant time difference between the signals of about $\Delta t \approx 0.02$ s. This time delay may be converted to a convective distance of $s \approx 0.3$ m using the free stream velocity as the convection velocity. This value is close to the sum of the distance of the three-hole probe head and the half chord length ($s = 0.13$ m $+ 0.1525$ m $= 0.2825$ m). Hence, this time difference is most probably created due to the location of the probe's head.

Considering the active flap case, Fig. 14 shows that the phase of the three-hole probe measurements is lower than in the fixed flap case. Furthermore, the differences increase towards higher frequencies. This is expected to result from the flap motion, which leads to additional added mass forces that in turn create a phase lead as seen in Fig. 11. As added mass effects are not included in this method, a decrease in phase is created.

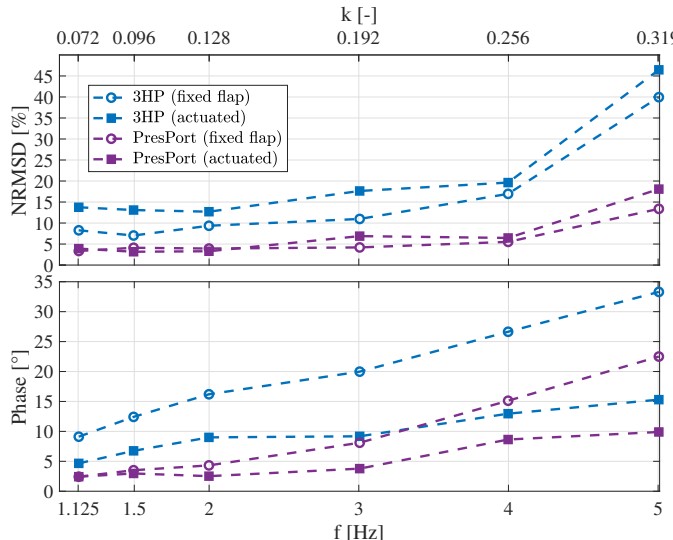

**Figure 14.** Normalized mean root square deviation and phase difference between lift estimation methods and full pressure distribution results.

Concerning the pressure port lift estimate, the error is substantially smaller for all frequencies and also the difference between the fixed and active flap cases is less significant. This is due to the fact that this method takes unsteady effects into account. Towards higher reduced frequencies the error increases, which is again expected to happen due to the neglected added mass effects within this method. Finally, the phase difference for the pressure port estimate is substantially smaller in comparison to the three-hole probe results. This is primarily due to the location of the ports on the airfoil. Again, for the fixed flap case a linearly increasing phase difference is present. Regarding the measurements for the active flap cases, the phase difference is




reduced and also the difference to the fixed flap case increases with raising frequency. Once more, this is due to the fact that additional added mass effects due to flap motion are not taken into account in the surface pressure port method.

## 7  Results of the load control

The presented lift estimation methods are employed as input to two separate feedforward control strategies. The performance
5  of two feedforward controllers is compared to a standard PID feedback controller in the current section.

In Fig. 15, ten seconds of the time series for the inflow disturbance at $k = 0.072$ is shown. Herein, the angle of attack and velocity variation, measured by the three-hole probe, are shown in the first and second diagram. In the third graph, the flap motion is shown and the resulting lift is depicted in the bottom diagram. The grey, dashed vertical line in all graphs indicates the start of the inflow grid motion. In these diagrams, the baseline case and the different controller strategies are plotted on top
10  of each other. For completeness, the time series for higher reduced frequency are given in appendix C.

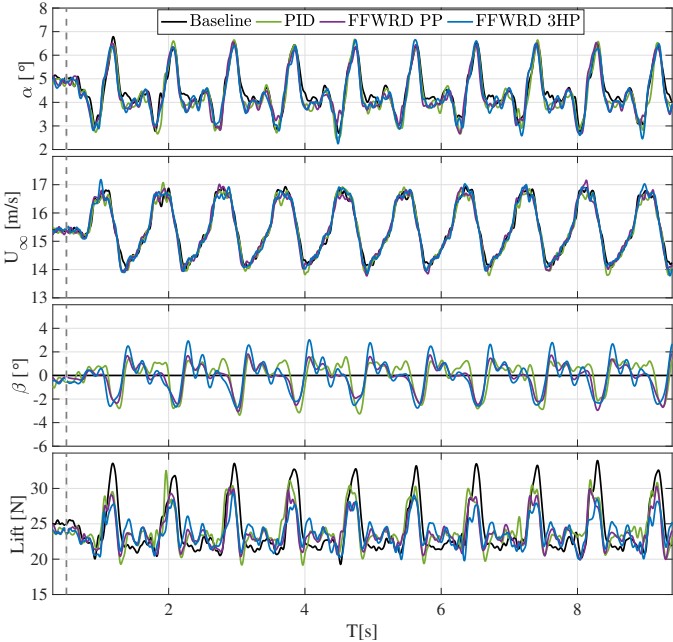

**Figure 15.** Time series of AoA and inflow velocity variation, flap motion and resulting lift amplitude at a reduced frequency of $k = 0.072$ for different control methods in comparison to uncontrolled baseline case.

As can be seen in the first two plots, the angle of attack and velocity variations, created by the inflow grid, are reproducible over time and between the different cases. Furthermore, the variations are periodic but not harmonic. The grid motion aimed at an in-phase disturbance between the angle of attack and velocity variation. The two properties are inherently changed by a moving grid in the wind tunnel: larger angles of attack leading to higher blockage, hence lower flow velocity. Therefore, an





independent harmonic disturbance of angle of attack and velocity could not be established at the employed frequencies. Such disturbances are for example seen in yaw case situations on wind turbines (in this case, angle of attack and velocity variation are $180°$ out-of-phase), (Bartholomay et al., 2018, Fig. 20).

Nonetheless, the inflow disturbances are employed here as test cases for the controller configurations and are considered more challenging than pure harmonic disturbances. Furthermore, the shape of the inflow disturbances also vary when the frequency is changing, as seen in appendix C. Therefore, each test case has to be analyzed separately. All quantitive results that will be explained in the following sections are given in Table E1.

## 7.1 Inflow disturbance at a reduced frequency of $k = 0.072$ ($f = 1.125$ Hz)

The bottom graph in Fig. 15 shows the resulting lift variation for the test case at $k = 0.072$. Distinct peaks are visible at the moment where the maximum angle of attack and the velocity maximum coincide (black line - baseline). The lift drops then to a fairly constant plateau before picking up again. Figure 15 (lower plot) clearly shows that all controller strategies accomplish a certain reduction of the lift maximum.

In order to analyze the success of the strategies in more detail, short term damage equivalent loads (DELs) are calculated. The comparison of the DELs can be seen in Fig. 16. In this graph, results are normalized by the baseline DEL, corresponding to the wing exposed to inflow disturbances with fixed flap at each analyzed frequency.

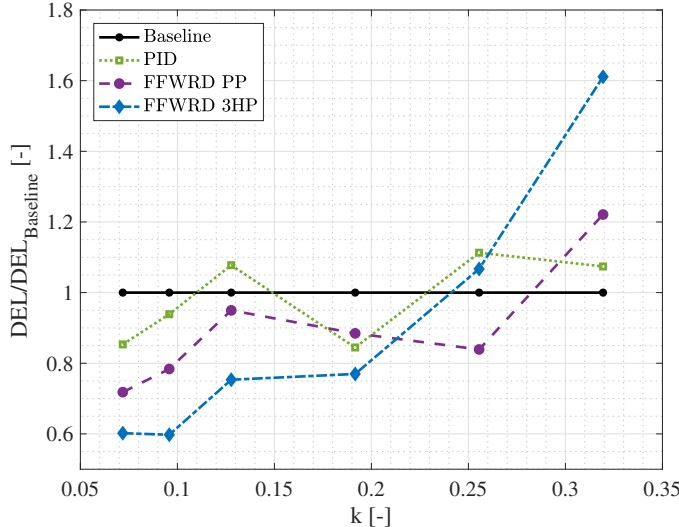

**Figure 16.** Comparison of short term damage equivalent loads for feedforward and feedback control.

At the reduced frequency of $k = 0.072$, the force balance based PID controller is capable to reduce the DEL to $85.33\%$. Employing the PP-feedforward controller reduces the DEL to $71.8\%$ and an even greater reduction is achieved by employing the 3HP-feedforward controller to a DEL of $60.2\%$. The lower performance of the PID controller is to be expected, as feedback





control has to experience a changing load first, before it can react to it. Feedforward control is capable to counteract the changing inflow conditions before they are felt by the wing. Despite the lift estimation based on the three-hole probe having a lower accuracy than the pressure surface based estimation (as seen in Fig. 14), the 3HP controller is superior in comparison to the other controllers. This indicates that in this test case the phase lead seen for the three-hole probe lift estimation is more
decisive for the success of the controller strategy than the inaccuracies of the lift estimation.

### 7.2   Inflow disturbance at a reduced frequency of $k = 0.096$ ($f = 1.5$ Hz)

At the reduced frequency of $k = 0.096$ the trends concerning the DELs are generally kept (Fig. 16). However, the success of the feedback control is lowered with a DEL of $93.9\%$ compared to the baseline. The PP-controller performance is also slightly lowered to $78.4\%$, whereas the 3-HP controller result remains almost constant at $59.8\%$. The lower alleviation level
of the feedback control and the PP-controller can be explained by the increasing challenge at higher frequencies of the inflow disturbance. This affects the 3-HP controller less, as the phase lead of the lift estimation increases and the error slightly decreases, see Fig. 14.

### 7.3   Inflow disturbance at a reduced frequency of $k = 0.128$ ($f = 2.0$ Hz)

At the reduced frequency of $k = 0.128$ the feedback controller fails to reduce the DEL and the PP-controller has a very small
positive effect on the DEL resulting in a level of $95.0\%$. In contrast, the 3-HP controller still reduces the fluctuation loads to a level of $75.3\%$ even though the load alleviation is not as good as seen for lower frequencies.

In order to understand the lowered performance of the different controller settings, it is worth looking at the (cumulative) power spectral density (CPSD) given in Fig. 17. As seen in this plot, the wing is not only affected by a disturbance at the base frequency of $k = 0.128$, corresponding to $f = 2$Hz, but also significantly by its first harmonic at a reduced frequency
of $k = 0.256$. The CPSD further shows that at the base frequency of $f = 2$ Hz all controllers successfully reduce the fluctuation in comparison to the baseline (see black line in lower plot Fig. 17). In agreement with the previous result at lower frequencies, the *3-HP*-controller is superior to the PP-controller and the feedback controller. At the first harmonic ($f = 4$ Hz) the step that each cumulative power spectral density experiences is quite conclusive. The baseline increase of the cumulative power spectral density at this frequency is $\Delta\mathrm{CPSD}_{f=4\mathrm{Hz,\ baseline}} = 27.9$ N$^2$, whereas the increase seen by the PID-controller
is $\Delta\mathrm{CPSD}_{f=4\mathrm{Hz,\ PID}} = 50.7$ N$^2$, $\Delta\mathrm{CPSD}_{f=4\mathrm{Hz,\ PP}} = 47.44$ N$^2$ for the PP-controller and $\Delta\mathrm{CPSD}_{f=2\mathrm{Hz,\ 3HP}} = 29.6$ N$^2$ for the 3HP-controller. This demonstrates that the 3HP controller has a comparable contribution to the overall loading as the baseline case at this frequency. On the contrary, the other controller types add significant loading at the first harmonic. Recalling that the calculation of DELs is based on rain-flow counting of cycles and weighting them by the Wöhler function, it is evident that a larger amplitude at the same frequency leads to more damage, hence more damage equivalent loading. As only the increase
in loading at the first harmonic ($f = 4$ Hz) for the 3HP controller is close to the baseline case, this controller strategy shows the lowest level of DELs (see blue line Fig. 16). It is expected that a pure harmonic disturbance would have been successfully alleviated by all controller configurations.

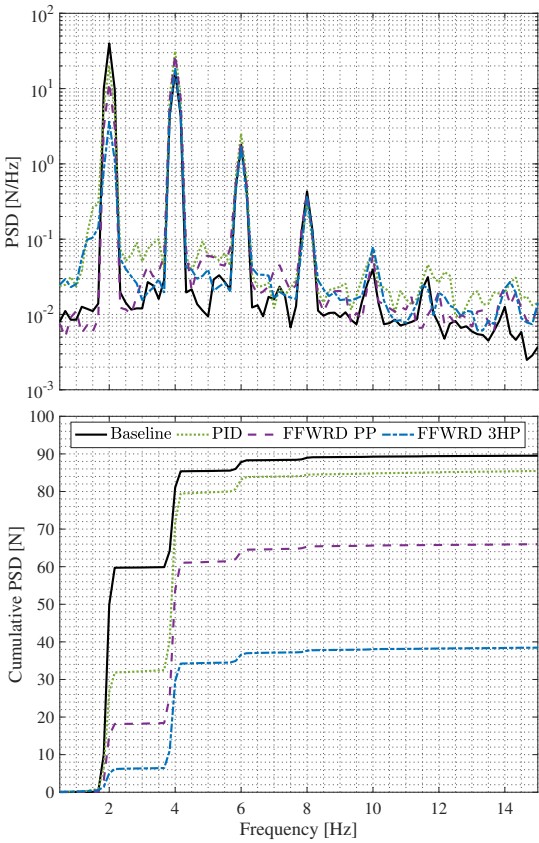

**Figure 17.** PSD and CPSD of the lift amplitude for the inflow disturbance with a base frequency of $k = 0.128$.

## 7.4 Inflow disturbance at a reduced frequency of $k = 0.192$ ($f = 3.0$ Hz)

Figure 16 shows the DELs at the reduced frequency of $k = 0.192$. It can be seen that the difference between the DELs of all studied cases decreases. The PID-controller achieves a reduction to $84.5\%$, the PP-controller $88.5\%$ and the 3HP-controller sees a slight increase in comparison to its previous result to a level of $77.0\%$.

5    Once more, in order to understand the performance of the different controller settings, it is worth looking at the CPSD, given in Figure 18. The result seen in terms of CPSD is comparable to the case seen for the test case at $k = 0.128$. In this particular test case, the disturbance at a frequency of $f = 3$ Hz leads to an increase in CPSD for all controller settings. Once more, the 3HP-controller is the most effective for this frequency, leading to an increase to $\mathrm{CPSD}_{\text{f=3Hz, 3HP}} = 25.7 \text{ N}^2$ which is $38\%$ of its baseline equivalent. Furthermore, the PP-controller reduces the load at this frequency to a level of $57\%$ and the PID-controller results

10   to $76\%$. This trend is not seen so clearly for the DELs, Figure 16, as the first harmonic ($f = 6Hz$) of the disturbance frequency has to be considered as well. Here, the PID-controller experiences the smallest step $\Delta\mathrm{CPSD}_{\text{f=6Hz, PID}} = 13.3 \text{ N}^2$, whereas the the two feedforward controllers see both an increase of $\Delta\mathrm{CPSD}_{\text{f=6Hz, PP}} = 21.2 \text{ N}^2$ and $\Delta\mathrm{CPSD}_{\text{f=6Hz, 3HP}} = 21.2 \text{ N}^2$.

In order to understand this effect, it is worth looking at the different phases introduced by the lift estimates (Fig. 14) and by the low-pass filter (Fig. 11). In particular, the latter introduces a phase shift of $\varphi_{\text{LP,f=6Hz}} = -155.3°$ with an additional phase shift due to the mechanics of the trailing edge flap of $\varphi_{\text{Flap,f=6Hz}} = -28.81°$ and the phase due to update time $\varphi = -21.6°$ resulting to a phase of $\varphi_{\text{LP+Mechanics,Update,f=6Hz}} = -205.7°$. This indicates that all controller fail to alleviate loads at this frequency for the current mechanical, controller and low-pass filter setup. It is expected that this effect is less severe for the PID-controller for multiple reasons: First of all, the frequency of 6Hz is beyond the phase turning point described in section 6.1, where the phase of the lift leads the flap motion due to added mass effects. The latter is seen by the force balance but not by the pressure based lift estimation methods. Thereby, a phase lead of the balance of $\Delta\varphi_{\text{f=6Hz}} = -21.9° - (-35.8°) = 13.9°$ is present in comparison to the pressure based methods. Secondly, as the force balance measurement is also affected by the force of the acceleration of the flap inertia, the change in lift due to flap motion is substantially less in comparison to pressure based methods. This can be seen at marker $A$ in Figure 11, where the gain of the lift amplitude is measured to $25\%$ by the force balance due to a $180°$ phase shift between the aerodynamic lift and the force created by the acceleration of the flap inertia. Hence, the pressure based methods estimate a substantially larger lift from the flap motion than what is actually measured in the force balance, leading to larger flap deflections and thereby to larger amplitudes at $f = 6$ Hz.

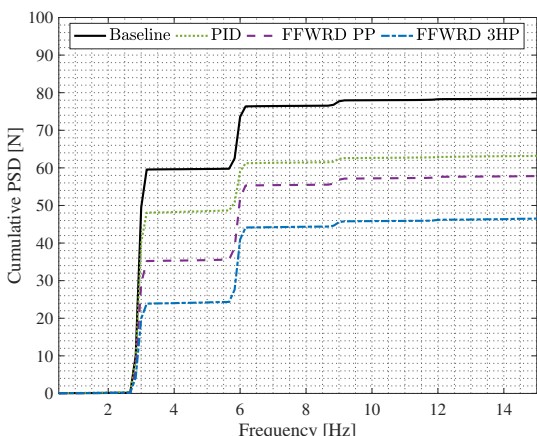

**Figure 18.** Cumulative power spectral density of the lift amplitude at the reduced frequency of $k = 0.192$.

The disturbances with frequencies of $k = 0.256$ and $k = 0.319$, corresponding to frequencies $f = 4$ Hz and $f = 5$ Hz respectively, are given for completeness to show where the controller options fail.

### 7.5 Inflow disturbance at a reduced frequency of $k = 0.256$ ($f = 4.0$ Hz)

Furthermore, Fig. 16 shows the DELs at a reduced frequency of $k = 0.256$. It can be seen in that the PID- and the 3HP-controller fail to lower the level of the damage equivalent load, whereas the PP-controller achieves a level of $84\%$ of its baseline equivalent. The corresponding cumulative power spectral density is given in Fig. 19, confirming that the PID- and the 3HP-controller already fail to alleviate the applied fluctuating load at $k = 0.256$. Additionally, the PP-controller has a slightly




higher CPSD level at $f = 4$Hz as the baseline, indicating that this controller is also not capable to alleviate the introduced load at this frequency.

This result is contradictory in comparison to the load alleviation seen at the first harmonic ($f = 4$ Hz) in the $k = 0.128$ case. This gives rise to two questions: First, why is the result of the 3HP-controller significantly worse, and second, why is the

PP-controller a lot better than what was seen previously at this frequency?

Concerning the failure of the 3HP controller, the phase lag introduced by the low-pass filter, update time and the flap mechanics, $\varphi_{LP+Mechanics+Update,4\text{Hz}} = -105° - 19.7° - 14.4° = -139.1°$, has to be taken into account, which leads to the poor load alleviation result. Furthermore, it is important to note that the error of the lift estimation of the three-hole probe starts to increase significantly at this frequency, Fig. 14. In particular, the NRMS of the three-hole probe estimate increases

to $19.7\%$, caused by a significant overestimation of the lift amplitudes as seen in the time series (see Fig. 13, right side, $k = 0.256$). It is expected that for the considered frequency of $f = 4$ Hz this effect is more significant for the $k = 0.256$ case than for the $k = 0.128$ case, as the amplitude of the lift fluctuation is $21.4\%$ larger ($L_{k=0.128,4\text{Hz}} = 2.8$ N in comparison to $L_{k=0.256,4\text{Hz}} = 3.4N$). This is reflected by an over-proportional increase of the flap motion amplitude of $\beta_{k=0.128,4Hz} = 1.7°$ by $70.6\%$ to $\beta_{k=0.256,4Hz} = 2.9°$.

In order to answer the second question, why the PP-controller shows a successful reduction of the DEL to a level of $83.9\%$ (see Fig. 16), the time series is considered first (see Fig. C4 lower plot). It can be observed that the resulting lift is changed in mean level, but the maximum peak to peak level is almost unaltered when comparing the PP-controller to the baseline result.

This is reflected in the cumulative power spectral density (see Fig. 19). The step at $f = 4Hz$, corresponding to the reduced frequency of $k = 0.256$, amounts to $\Delta\text{CPSD}_{f=4\text{Hz},PP} = 37.5$ N$^2$ for the PP-controller which is slightly larger in comparison

to the baseline case ($\Delta\text{CPSD}_{f=4\text{Hz},Baseline} = 36.3$ N$^2$). Hence, the PP-controller does not successfully alleviate the loading at $f = 4$Hz.

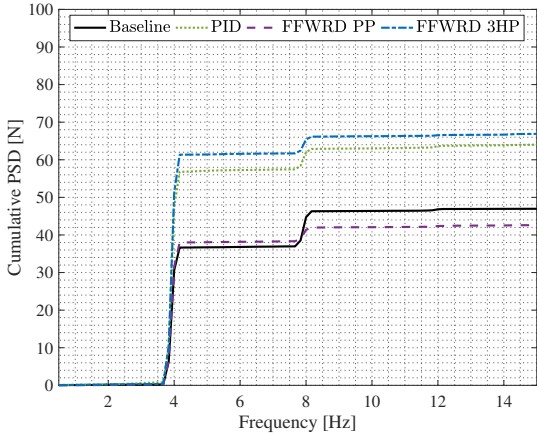

**Figure 19.** Cumulative power spectral density of the lift amplitude at the reduced frequency of $k = 0.256$.



As a side-note, when the signals are low-pass filtered at $f = 6$ Hz, thus, including the disturbance frequency, but cutting the first harmonic at $f = 8$ Hz out, and then calculating the DELs, the result of the failing PP-controller is confirmed. However, when including the first harmonic at $f = 8$ Hz, the DEL is smaller for the PP-controller. Therefore, the loading at $f = 8$ Hz has to be considered as well. As seen in Fig. 19, the step at this frequency for the PP-controller ($\Delta\text{CPSD}_{\text{f=8Hz, PP}} = 3.6$ N$^2$) is

substantially smaller than for the baseline case ($\Delta\text{CPSD}_{\text{f=8Hz, Baseline}} = 9.3$ N$^2$). This leads to the apparent lower DEL result seen in Fig. 16 and also to a lower CPSD level for the PP-controller (see Fig. 19). It is expected that the lower step in CPSD at $f = 8$Hz is caused by the configuration of this particular test rig and the employed controller configuration. As seen in Fig. 11, above a frequency of $f = 7$ Hz the lift response due to flap motion is inverted due to the force that is created by the acceleration of the flap inertia. This creates a phase shift for the lift response due to flap motion of $\varphi_{\text{L}_{\text{Balance}}} = 180°$. Taking into account

the large phase due to low-pass filter, mechanics and update time $\varphi_{\text{LP+Mechanics+Update,8Hz}} = -195° - 36.6° - 28.8° = -260.4°$, these two effect partially cancel each other out, leading to a certain improvement of the 8Hz disturbance. As these frequencies are out of the scope of this paper, they were not further analyzed.

### 7.6    Inflow disturbance at a reduced frequency of $k = 0.319$ ($f = 5.0$ Hz)

The final test case is given by an inflow disturbance with a reduced frequency of $k = 0.319$. Figure 16 shows that both feed-

forward strategies fail to lower the fluctuating load. This can be explained by a significant overestimation of the lift amplitude by the pressure based lift estimation methods (see Fig. 13 right side, fifth row). This effect cumulates to a NRMS error of the three-hole probe measurement, Fig. 14, of $\text{NRMS}_{\text{3HP}} = 46\%$ and $\text{NRMS}_{\text{PP}} = 19\%$ for the surface pressure method. Additionally, the introduced phase lag of the low-pass filter and the flap mechanics at $f = 5$ Hz amount to $\varphi_{\text{LP+Mechanics+Update,5Hz}} = -130° - 24.3° - 18.0° = -172.3°$ indicating that the flap motion is almost completely in phase with the lift amplitude. This

implies that a positive flap down motion - creating higher lift - is almost in phase with a positive lift amplitude affecting the wing. This can be confirmed when looking at the flap position and lift amplitude given in Figure C5, third and forth row.

## 8    Summary and Conclusion

In this study feedforward control strategies for wind turbine applications to alleviate fluctuating loads on a two-dimensional wing are presented. Three-hole probe or surface pressure port lift estimates served as input for the feedforward control. Within

this paper, the lift estimation methods were presented first and then its application to load control was shown.

In a first step, test cases without inflow disturbance and only flap motion were analyzed. The surface pressure port and the three-hole probe based estimate agree fairly well with the balance measurements and with the lift that is based on the integration of the complete pressure port set ($L_{\text{FULLPP}}$) up to reduced frequency of $k = 0.192$. For higher frequencies the amplitude measured by the balance drops due to the mechanical properties of the current setup. The pressure port lift estimate

still compares well to the $L_{\text{FULLPP}}$, whereas the three-hole-probe estimate overpredicts the gain as unsteady effects are not captured. Furthermore, it is seen that at $k = 0.318$ the phase between flap motion and lift response turns. This effect is not captured by the two presented methods as added mass effects are not incorporated.





Regarding the test cases with inflow disturbance and fixed flap, both methods show decent results for the wing exposed to reduced frequency of up to $k = 0.192$. The pressure port lift estimate is generally significantly better than the three-hole probe estimate, as unsteady effects are captured. The latter method overpredicts significantly the lift amplitude. This effect worsens with increasing frequency, seen already at $k = 0.255$ and leading to unacceptable results at $k = 0.319$. Regarding cases where
the flap of the wing is additionally actuated, the error of both methods is further increased, whereby the three-hole probe method is more affected. Regarding the phase of both methods, the three-hole probe method is advantageous, especially as the probes head is located upstream of the wing. However, when the flap is active, the phase lead is halved for the three-hole probe method.

The load control results show that the presented pressure based controllers improve the load alleviation capability of the
present trailing edge flap setup significantly up to a frequency of $k = 0.192$. Even though, the three-hole probe based lift estimate is worse in quality in comparison to the surface pressure port estimate, the phase advantage leads to significant superiority. Moreover, it is expected that in a setup where a pure sinusoidal disturbance can be created, a higher load alleviation potential can be achieved. In the current setup, higher harmonics increase the challenge for the controller setup. Above a frequency of $k = 0.192$ all controller types fail. It is expected that the use of a low-pass filter with increased cut-off frequency
or a bandpass filter can further improve the results. Furthermore, creating an unsteady lift estimation method based on the three-hole probe will combine a high-quality lift estimation with an advantageous phase lead.

The current study presents approaches for lift estimation that may serve as input for load controllers on research or industry scale turbines. In particular employing surface pressure measurements that do not rely on tubing will be beneficial as they avoid issues such as clogging by insects or dust. Moreover, trailing edge flaps are a promising device for the alleviation of fatigue
and extreme loads of wind turbines.

The authors aim at transferring the results to experiments on the Berlin Research Turbine. It is expected that feedforward control will significantly improve load reduction capabilities of the setup. Furthermore, it also planned to employ the measurement of the blade root bending moment of the preceding blade as an input to the controller as this will allow for an additional phase lead which was shown to be very decisive in this study.

*Competing interests.*   The authors declare that they have no conflict of interest

*Acknowledgements.*   This study was funded by the Germany Research Foundation (DFG) within the framework of the PAK 780II project.



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





## Appendix A: Nomenclature

|  | Unit | Meaning |
|---|---|---|
| $A, b$ | — | indicial constants of the ATEFlap Model |
| $\alpha$, AoA | ° | angle of attack |
| $\alpha_{\text{eff}}$ | ° | effective angle of attack |
| $\alpha_{\text{probe}}$ | ° | angle between the flow and the probe axis |
| $\alpha_{\text{probe,section}}$ | ° | flow angle at the probe head in reference to the chord |
| $\alpha_{\text{turntable}}$ | ° | geometric angle of attack set by the turntable |
| $\beta$ | ° | flap angle |
| $\beta_{\text{SP}}$ | ° | setpoint of the flap angle |
| $c$ | m | chord of the wing |
| $c_{\text{grid}}$ | mm | chord of the grid |
| $C_{\text{L}}, C_{\text{D}}, C_{\text{M}}$ | — | lift, drag and moment coefficient |
| $c_{\text{p}}$ | — | pressure coefficient |
| CPSD | N | cumulative power spectral density |
| $D$ | — | total damage |
| $d_{\text{i}}$ | — | damage at different loading |
| DEL | — | damage equivalent load |
| $f$ | Hz | frequency |
| $f_{\text{actuation}}$ | Hz | actuation frequency |
| $f_{\text{cut-off}}$ | Hz | low-pass cut-off frequency |
| $F^{*}_{dydxLE}, H_{y,i}, H_{dyd\epsilon,i},$ | — | deflection shape integrals of the ATEFlap model |
| $f_{\text{eq}}$ | Hz | equivalent frequency |
| $f_{\text{s}}$ | Hz | sampling frequency |
| $f_{\text{update}}$ | Hz | update frequency of the controller routine |
| $g_{\beta}$ | — | pressure difference coefficient due to the deflection of the trailing edge flap |
| $g_{\text{c}}$ | — | circulatory pressure difference distribution |
| $g_{\text{camber}}$ | — | pressure difference coefficient due the camber of the airfoil |
| $h$ | m | height of the windtunnel |
| $k$ | - | reduced frequency |
| $K_1, K_2, K_3$ | - | calibration constants |
| $k_{\text{c}}$ | - | steady lift slope |



| | Unit | Meaning |
|---|---|---|
| $k_{\dot{\alpha}}$ | - | coefficient due to added mass of the airfoil motion |
| $K_\mathrm{p}$ | - | proportional gain of the PID controller |
| $L$ | N | lift |
| $L_\mathrm{3HP}$ | N | lift estimate based on the three-hole probe |
| $L_\mathrm{FULLPP}$ | N | lift measurement based on the surface pressure ports |
| $L_\mathrm{PP}$ | N | lift estimate based on three selected surface pressure ports |
| $L_\mathrm{ref}$ | N | reference lift |
| $m$ | — | Wöhler exponent |
| $N_\mathrm{i}$ | — | number of cycles to failure at different loading |
| NRMS | — | normalized mean root square deviation |
| $p$ | Pa | pressure |
| $p_{p,LE,SS12.5\%}$ | Pa | pressure difference between the leading edge and the port at $x/c = 12.5\%$ on the suction side |
| $p_{p,LE,PS12.5\%}$ | Pa | pressure difference between the leading edge and the port at $x/c = 12.5\%$ on the suction side |
| $p$ | — | order of rotation |
| $\varphi$ | ° | phase |
| $\rho$ | $\mathrm{kg \cdot m^{-3}}$ | density |
| $Re$ | — | Reynolds number |
| $s$ | m | span of the wing |
| $s_\mathrm{grid}$ | mm | span of the grid |
| $t, T$ | s | time |
| $T_\mathrm{i}$ | s | integration time of the PID controller |
| $T_\mathrm{d}$ | s | derivative time of the PID controller |
| $U_\infty$ | — | inflow velocity |
| $U_\mathrm{rel}$ | — | relative velocity |
| $w$ | m | width of the windtunnel |
| $x, y, z$ | m | coordinates |
| $x_\mathrm{probe}, y_\mathrm{probe}$ | m | coordinates in reference to the probe head |
| $x_\mathrm{probe,section}, y_\mathrm{probe,section}$ | m | coordinates in reference to the airfoil chord |





## Appendix B: Lookup Table for the Estimation of the Inflow Velocity

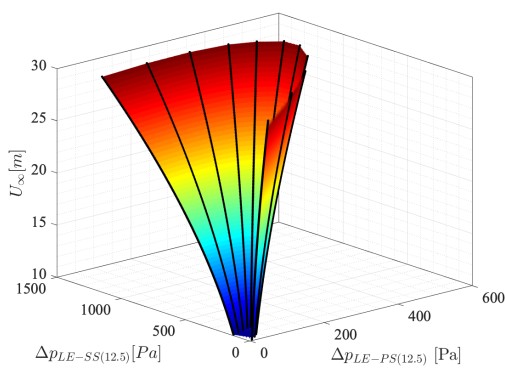

**Figure B1.** Calibration function for estimating the inflow velocity. Example shows the result for a flap angle of $\beta = 0°$

## Appendix C: Time Series

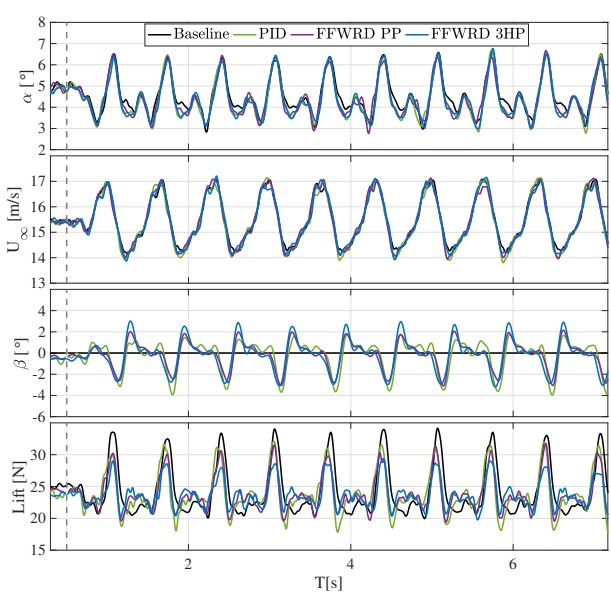

**Figure C1.** Time series of angle of attack and velocity variation, flap motion and resulting lift amplitude at a reduced frequency of $k = 0.096$.



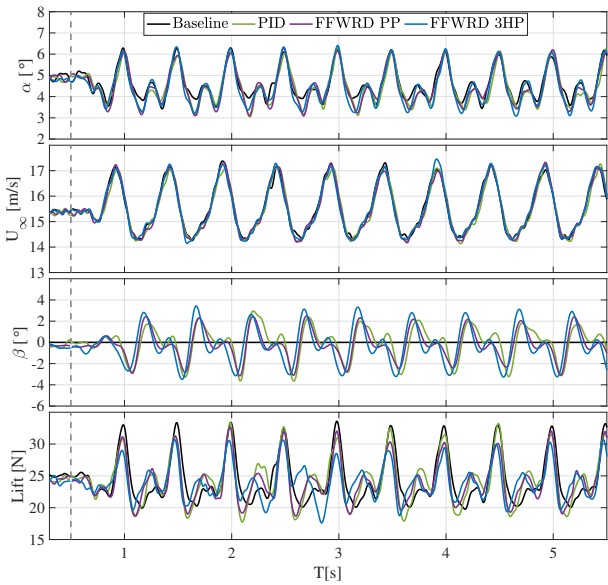

**Figure C2.** Time series of angle of attack and velocity variation, flap motion and resulting lift amplitude at a reduced frequency of $k = 0.128$.

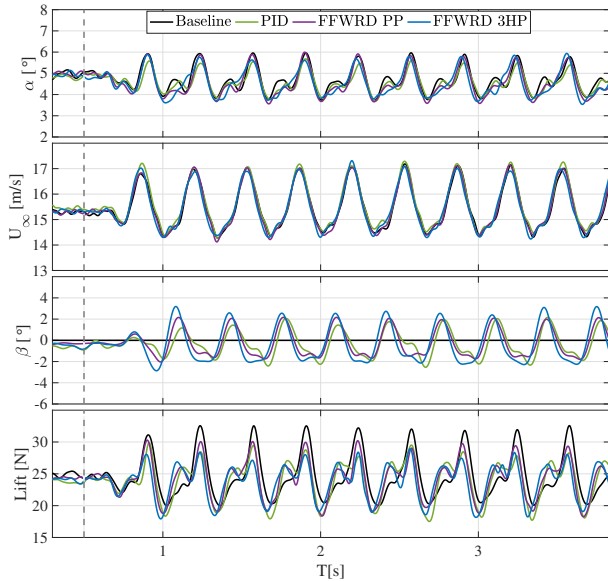

**Figure C3.** Time series of angle of attack and velocity variation, flap motion and resulting lift amplitude at a reduced frequency of $k = 0.192$.



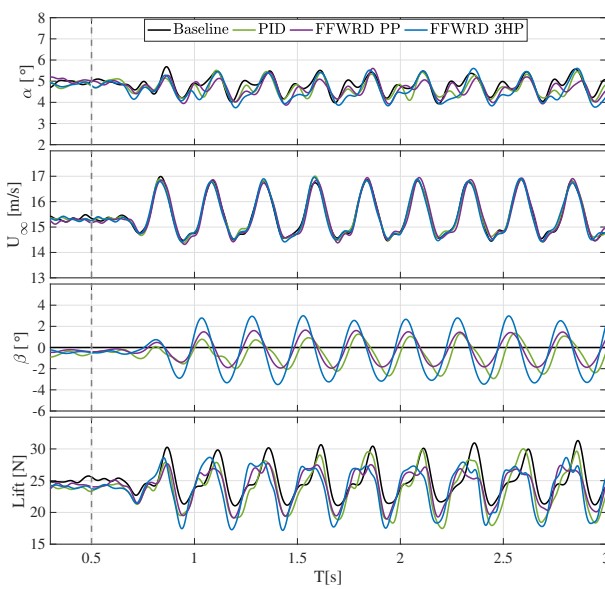

**Figure C4.** Time series of angle of attack and velocity variation, flap motion and resulting lift amplitude at a reduced frequency of $k = 0.256$.

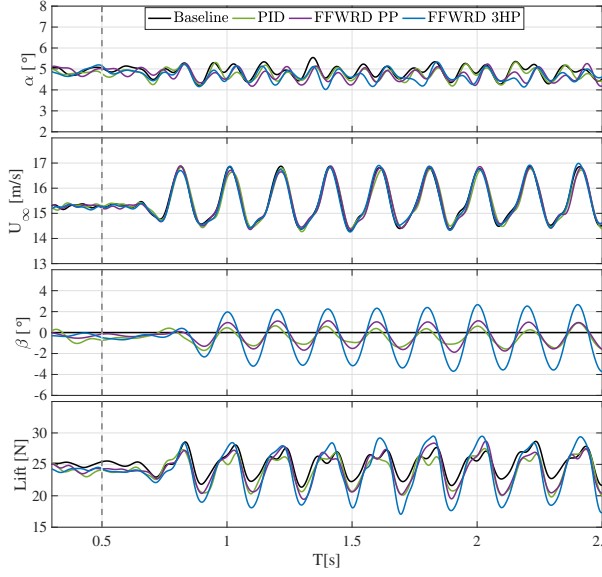

**Figure C5.** Time series of angle of attack and velocity variation, flap motion and resulting lift amplitude at a reduced frequency of $k = 0.319$.



## Appendix D: Power Spectral Density

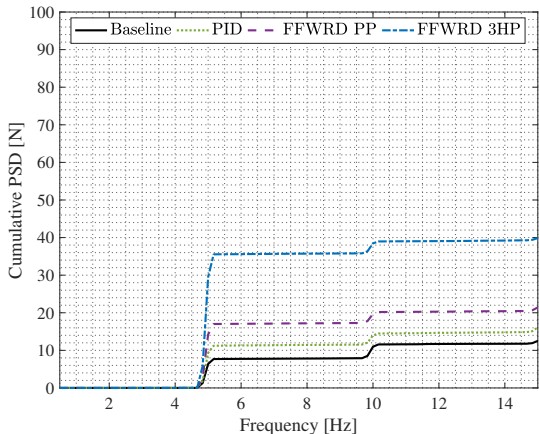

**Figure D1.** Cumulative power spectral density of the lift amplitude at the reduced frequency of $k = 0.319$.



## Appendix E: Summary of the load alleviation results

**Table E1.** Summary of the load alleviation results. $\Delta CPSD_{base}[\mathrm{N}^2]$ denotes the increase of cumulative spectral density at the base frequency. Accordingly, $\Delta CPSD_{1st}[\mathrm{N}^2]$ corresponds to the increase at the first harmonic. Green color indicates a load alleviation in comparison to the baseline, red corresponds to an increase in loading. Black indicates no change.

| $k[-]$ | | DEL[N] | $\Delta CPSD_{base}[\mathrm{N}^2]$ | $\Delta CPSD_{1st}[\mathrm{N}^2]$ |
|---|---|---|---|---|
| | **Baseline** | **13.5** | **44.9** | **27.2** |
| | *PID* | 11.6 | 14.4 | 16.7 |
| 0.072 | *PP* | 9.7 | 8.4 | 15.1 |
| | *3HP* | 8.2 | 6.9 | 6.8 |
| | **Baseline** | **14.6** | **60.5** | **30.0** |
| | *PID* | 13.7 | 20.21 | 30.85 |
| 0.096 | *PP* | 11.5 | 11.2 | 23.7 |
| | *3HP* | 8.7 | 5.4 | 11.6 |
| | **Baseline** | **14.4** | **59.6** | **25.5** |
| | *PID* | 15.5 | 31.3 | 47.1 |
| 0.128 | *PP* | 13.6 | 18.1 | 42.7 |
| | *3HP* | 10.8 | 5.8 | 27.8 |
| | **Baseline** | **13.7** | **59.4** | **16.6** |
| | *PID* | 11.6 | 47.8 | 12.61 |
| 0.192 | *PP* | 12.1 | 35.1 | 19.8 |
| | *3HP* | 10.5 | 23.7 | 19.9 |
| | **Baseline** | **10.6** | **36.4** | **9.4** |
| | *PID* | 11.8 | 56.1 | 5.5 |
| 0.256 | *PP* | 8.9 | 37.8 | 3.7 |
| | *3HP* | 11.3 | 61.0 | 4.5 |
| | **Baseline** | **5.8** | **7.7** | **3.7** |
| | *PID* | 6.2 | 11.2 | 3.0 |
| 0.319 | *PP* | 9.7 | 7.0 | 2.9 |
| | *3HP* | 12.4 | 9.3 | 3.2 |