# Peer review of "Pressure Based Lift Estimation and its Application to Feedforward Load Control employing Trailing Edge Flaps"

_Wind Energy Science, 2020_

## Referee Comment (RC1) · Anonymous Referee #1 · 13 Oct 2020

The paper describes the implementation of a trailing edge flap controller for a quasi two-dimensional airfoil interacting with two-dimensional inflow disturbance created by an active grids of airfoil profiles. Significant room is dedicated to validation of lift estimation strategies by using a three-hole probe before the airfoil and by using three pressure ports on the surface of the airfoil. By comparing with a PID controller, it is shown that the proposed lift estimation strategies perform better at lower reduced frequencies and are suitable to alleviate damage equivalent loads by active trailing edge flap control.

In general, this is a very well written paper that reports on interesting ideas and a large

body of work. There is little to critique.

For wind turbine applications, it would be useful if the authors could elaborate a bit further which regime, i.e., which exact reduced frequencies, etc. would be expected for turbines of typical industrial rotor diameter and also the Berlin research scale turbine (BeRT) that seems to be the primary target for this load controller.

Some minor typos and questions: p.2, 29: . . .arises which sensor input. . . p.4, 18: . . .. by a two-dimensional active grid . . . p.6, 31: The underlying methodology . . . p.6, 33: The reference static pressure was obtained from . . . p.8, 5: . . . Fourier transformed . . ., 7: . . .Fourier transform p.16, 9: . . .which is chosen as. . . p.16, 10 . . . equation 11. . . , 13: Eq. 12 . . . Make sure equations are referenced consistently. p.17, 9: What exactly are A and b? Not referred to in any given equations.

---

## Referee Comment (RC2) · Anonymous Referee #2 · 6 Nov 2020

[referee-annotated manuscript omitted]

---

## Author Comment (AC1) · 13 Dec 2020

Please see the supplement PDF.

Please also note the supplement to this comment:
https://wes.copernicus.org/preprints/wes-2020-91/wes-2020-91-AC1-supplement.pdf

---

## Author Comment (AC2) · 13 Dec 2020

**Response to Reviewer 2**

Title: Pressure Based Lift Estimation and its Application to Feedforward Load Control employing Trailing Edge Flaps

**Dear Reviewer,**

We thank you for your interest in our work and the helpful suggestions to improve our paper. The summary of your review is:

**Summary of the review, uploaded 06.11.2020:**

Generally a good article, with insights on the mechanisms of utilizing pressure based lift estimation for load control with active flaps. More details on many of the aspects of the experiments and modeling can improve the quality of the article. Detailed comments are included in the uploaded pdf file.

Please see our point by point answer to your comments in the following. Furthermore, changes in the manuscript according to your comments are marked in blue.

**Page 1**

L1: Shouldn't the title reflect also the fact that wind tunnel testing is involved?

Generally, we agree to your comment. However, due to the lengthiness of the title:

Pressure Based Lift Estimation and its Application to Feedforward Load Control employing Trailing Edge Flaps

it is decided to keep the title as is. The interested reader is referred to the abstract where more information about the paper is given.

L15: The concept of reducing LCOE with active load control is mostly driven by the fact that larger rotors (+AEP) can be produced for the same loading. This should be clear when mentioning cost saving from material.

We agree to your comment that active load control is employed to design larger rotors (+AEP) for the same loading. Additionally, active load control can be used to reduce loads on a turbine and thereby reducing the capital expenditure as the less material is needed. Both ideas are added to the manuscript.

L19: Pitch bearing damage might be more important than response time.

Generally, the goal of any active load control method is to reduce loads. It is implicitly understood that there is an actuation cost in order to achieve this goal. Actuation wear will occur in both pitch and TE flap actuators and cannot be regarded as a disadvantage for one type of actuator only. As an additional disadvantage of the pitch actuators, we added the fact that it cannot effectively mitigate local blade loading.

**Page 2**

L21: This statement, which is afterwards used as an argument for the present study, is quite confusing. Probably rephrasing it to reflect the challenges/limitations of field testing will clarify this (if that's the intention of the statement).

We have changed the concerning paragraph to clarify the argument.

**Page 3**

L13 Few pressure taps compared to mounted probes seem like a more feasible option considering realistic application (installation, vibration, drag). Nevertheless pressure taps also potentially face the same clogging risk

(which is more due to moisture and not particles considering <0.5mm holes). Maybe the argumentation should be more clear in the text.

Thanks for your comment.

- 1) We have added 'moisture' in the manuscript.
- 2) And yes, pressure taps might suffer the same clogging risk as mounted probes. Nonetheless, for a lab scale experiments they are generally the choice for surface pressure measurements. For full scale applications thin film surface pressure sensors might be more feasible. This idea was added to the manuscript.

**L32 This is pretty low. You should at least comment of the validity of the cases compared to full scale.**

Conducting experiments on research scale turbines generally leads to the issue of low Reynolds numbers. Please compare to the Oldenburg Turbine (https://www.mdpi.com/1996-1073/12/7/1306/htm - here the maximum is Re=120k), Milano Turbine (https://iopscience.iop.org/article/10.1088/1742-6596/524/1/012061/pdf) where the Reynolds number is scaled by a factor 225 to the Innwind Reference Turbine. Herein, they argue, that Reynolds scaling was conducted trying to find a compromise between low Reynolds number and an excessive control bandwidth. This statement was added to the manuscript.

**Page 4**

L3 I guess you mean 1p (1/rev), it's confusing using 'pi'.

We have added (1/revolution) to clarify this statement.

**Page 5**

**L5 Is this angle used in all comparisons? What about wind tunnel corrections for blockage, flow curvature etc?**

We agree that a correction for solid body and wake blockage as well as stream line curvature is generally possible for this setup. However, the aim of the current paper is to compare different lift estimation methods between each other, which are partially calibrated on the underlying steady polars. Therefore, the fundamental outcome of the study is not changed as the correction would be propagated into the lift estimation methods. Nonetheless, if the interested reader is aiming for a CFD comparison all necessary parameters for blockage correction (chord, airfoil type, thickness, span, tunnel dimensions) are given in the paper.

Nonetheless, your point is taken, and the correction was calculated according to Barlow (Low Speed Wind Tunnel Testing 3rd ed, 1999). The result shown in the following figure, where the solid lines correspond to the corrected results. It can be seen that the difference in the linear region and for small flap angles is small, increasing towards larger angles of attack. However, as all following plots in the paper are grounded on the uncorrected values, this plot is not incorporated into the new manuscript in order not to confuse the reader.

Figure 1 Comparison between uncorrected balance measurements (dashed lines) and blockage corrected results (solid lines).

**Page 6**

**L17 How is the effect of the arm and all other disturbances taken into account when comparing to an undisturbed airfoil?**

Generally, the effect of the lever arm and the three hole probe (+ its support) is not taken into account and a conclusive answer can only be given by conducting a full 3d CFD study. However, the effect of the lever arm is expected to be negligible as the spanwise extension is less than 1%. Regarding the three hole probe, there might be a more significant effect due to the extension of the support of the probe. Nonetheless, as the spanwise extension is less than 3.6% the effect is expected to be very small.

L19 This is the first time pressure taps are mentioned in the text. More detail is needed (when mentioned again afterwards): positions, size. Moreover, usually pressure taps are not located in the same spanwise position in order to avoid disturbances)

We agree to your comments.

- 1) The coordinates of the pressure taps were added in a table in the appendix. Also the diameter of the holes is added to the manuscript.
- 2) Spanwise Positions: It is agreed that a varying spanwise position, which was not conducted due to manufacturing reasons, would have been beneficial for the results. However, a comparison of the cp distribution to xfoil calculations yields satisfying results. This is expected due to the fairly large spacing between the pressure taps.

Figure 2 Comparison of measured cp distributions to XFOIL calculations. Exemplary alpha= $5^{\circ}$  for three flap angles (- $5^{\circ}$ ,  $0^{\circ}$ ,  $5^{\circ}$ ) are shown.

**Page 9**

L2 It is very important when presenting these plots to verify how accurate the measured polars are compared to previous tests/CFD/even xFoil, especially considering the added disturbances of the setup. Moreover, the geometric angles of the turntable should not be used without including tunnel corrections. Is this the case? A steady state comparison of force balance and pressure taps should also be presented at some point, since the two methods are compared afterwards.

We agree to your comment. As this paper concentrates on lift estimation (and comparison) Figure 5 (polars of lift, drag and moment) of the original manuscript is exchanged for a figure presenting lift only. Herein, the comparison between balance measurements and lift calculations based on the integration of all pressure ports are shown. Furthermore, a comparison to XFOIL is shown for a flap angle of beta =  $0^{\circ}$ . Additional XFOIL polars for different flaps angles are not added to keep the plot readable. The accompanying description of Figure 5 is changed in the manuscript.

---

## Author Comment (AC3) · 13 Dec 2020

**Response to Reviewer 1**

Title: *Pressure Based Lift Estimation and its Application to Feedforward Load Control employing Trailing Edge Flaps*

Dear Reviewer,

We thank you for your interest in our work and the helpful suggestions to improve our paper. The summary of your review is:

**Summary of the review, uploaded 13.10.2020:**

*The paper describes the implementation of a trailing edge flap controller for a quasi two-dimensional airfoil interacting with two-dimensional inflow disturbance created by an active grids of airfoil profiles. Significant room is dedicated to validation of lift estimation strategies by using a three-hole probe before the airfoil and by using three pressure ports on the surface of the airfoil. By comparing with a PID controller, it is shown that the proposed lift estimation strategies perform better at lower reduced fre- quencies and are suitable to alleviate damage equivalent loads by active trailing edge flap control.*

*In general, this is a very well written paper that reports on interesting ideas and a large body of work. There is little to critique.*

Furthermore, please see our point by point answer to your comments in the following. Moreover, changes in the manuscript according to your comments are marked in magenta.

**1) Comment**

For wind turbine applications, it would be useful if the authors could elaborate a bit further which regime, i.e., which exact reduced frequencies, etc. would be expected for turbines of typical industrial rotor diameter and also the Berlin research scale turbine (BeRT) that seems to be the primary target for this load controller.

For a comparison the 75% spanwise position of the DTU 10 MW reference turbine at rated conditions was chosen. The reduced frequencies for 1p and 3p disturbances are 0.019 and 0.056, respectively. In comparison to the Berlin research turbine where at the same spanwise position for 1p and 3p disturbances are k=0.09 and 0.25, respectively. This information is added in section 2. Experimental Setup.

| DTU 10 MW | | | BeRT | | |
|---|---|---|---|---|---|
| 0.75 Radius | 67,98214 | m | 0.75 Radius | 1,125 | |
| Chord at 0.7 radius | 2,59 | m | Chord at 0.7 radius | 0,2 | |
| Rot Freq | 0,16 | Hz | Rot Freq | 3 | |
| Wind | 12 | m/s | Wind | 6,5 | |
| V_inplane | 68,34305 | m/s | V_inplane | 21,20573 | |
| Vrel | 69,38856 | m/s | Vrel | 22,17956 | |
| | | | | | |
| 1p Freq | 0,16 | Hz | 1p Freq | 3 | Hz |
| 3p Freq | 0,48 | Hz | 3p Freq | 9 | Hz |
| | | | | | |
| | | | | | |
| Red. Freq 1p | 0,018762 | | Red. Freq 1p | 0,084986 | |
| Red. Freq 3p | 0,056286 | | Red. Freq 3p | 0,254958 | |

**2) Some minor typos and questions**

p.2, 29: . . .arises which sensor input. . . – 'on' deleted

p.4, 18: . . .. by a two-dimensional active grid . . . 'an' exchanged by 'a'

p.6, 31: The underlying methodology . . . 'underlaying' exchanged by 'underlying'

p.6, 33: The reference static pressure was obtained from . . . 'used' exchanged by 'obtained'

p.8, 5: . . . Fourier transformed . . ., 7: ...Fourier transform . 'fourier' exchanged by 'Fourier'

p.16, 9: ...which is chosen as... 'as' added

p.16, 10 ... equation 11... , 13: Eq. 12 . . . Make sure equations are referenced consistently.
'equation' exchanged by 'Eq.'

p.17, 9: What exactly are A and b? Not referred to in any given equations.

A and b are constants that account for non-zero thickness of airfoils in the indicial unsteady aerodynamics code
of Gaunaa and Bergami. Therefore, the sentence: 'In order to account for the non-zero thickness of the airfoil,
the indicial constants for the model are calculated in reference to Bergami et al. (2013) and slightly adjusted: A =
[0.389; 0.264] and b = [0.380; 0.0564].' describes in the authors view what the constants mean. The interested
reader is referred to the detailed documentation of the ATEFlap Model referenced in the paper.

---

## Author Response (AR2)

**Response to Gerard van Bussel**

Title: *Pressure Based Lift Estimation and its Application to Feedforward Load Control employing Trailing Edge Flaps*

Dear Gerard van Bussel,

We thank you for accepting our paper and for your comment on Figure 4. We agree, that the scale and view in this plot was not too useful. We presented the plot this way, as we wanted to point onto the resonance peaks. In order to clarify the diagram, we changed the plot from greyscale to colour and we show the diagram in an isometric view now.

[Figure]

**Figure 4.** Waterfall diagram of the moment and normal force at different actuation frequencies for the trailing edge flap(y-axis). Experiment was conducted with the tunnel speed set to $u_\infty = 15\text{m/s}$

Furthermore, we agree to your point that the diagram is very comparable to a Campbell diagram, but it strictly speaking isn't, as you point out. Nonetheless, we have added this point in accompanying paragraph as this might help to reader to understand the plot faster:

**2.4 Frequency analysis of the test rig**

The study presented in this paper is considered an aerodynamic and not an aeroelastic experiment. Therefore, the test-rig was analyzed for its structural eigenfrequencies. The excitation for this task was driven by the trailing edge flap. Multiple runs at various fixed frequencies were conducted and a time series of the force and torque balance were measured for each run. Each time series was Fourier transformed and the results were stacked, yielding a waterfall diagram (Fig. 4). The diagram is comparable to a Campbell diagram, whereas for the current test rig the flap motion is used for excitation and not a rotational frequency as done for rotating machines. In Fig. 4 the result for the measurement of the moment is shown on the left, the normal force is depicted on the right side. The $y$-axis depicts the set actuation frequencies and in the $x$ axis corresponds to the Fourier transform of the signal.  Diagonal lines represent the actuation frequency and its multiples, with the left most line corresponding to the actuation frequency. As can be seen for the moment, there is a significant response at 16.6Hz, which is expected to be the torsional eigenfrequency of the test rig. This frequency appears also in the plot for the normal force. Additionally, a strong response can be seen at 24.8Hz which is expected to correspond to the normal eigenfrequency.

Thank you very much and kind regards,

Sirko Bartholomay